# Off-Policy Risk Assessment in Contextual Bandits

**Audrey Huang**
Department of Computer Science
University of Illinois Urbana-Champaign
Urbana, IL 61801
audreyh5@illinois.edu

**Liu Leqi**
Machine Learning Department
Carnegie Mellon University
Pittsburgh, PA 15213
leqil@cs.cmu.edu

**Zachary C. Lipton**
Machine Learning Department
Carnegie Mellon University
Pittsburgh, PA 15213
zlipton@cmu.edu

**Kamyar Azizzadenesheli**
Department of Computer Science
Purdue University
West Lafayette, IN 47907
kamyar@purdue.edu

## Abstract

Even when unable to run experiments, practitioners can evaluate prospective policies, using previously logged data. However, while the bandits literature has adopted a diverse set of objectives, most research on *off-policy evaluation* to date focuses on the expected reward. In this paper, we introduce Lipschitz risk functionals, a broad class of objectives that subsumes conditional value-at-risk (CVaR), variance, mean-variance, many distorted risks, and CPT risks, among others. We propose *Off-Policy Risk Assessment* (OPRA), a framework that first estimates a target policy's CDF and then generates plugin estimates for any collection of Lipschitz risks, providing finite sample guarantees that hold simultaneously over the entire class. We instantiate OPRA with both importance sampling and doubly robust estimators. Our primary theoretical contributions are (i) the first uniform concentration inequalities for both CDF estimators in contextual bandits and (ii) error bounds on our Lipschitz risk estimates, which all converge at a rate of $O(1/\sqrt{n})$.

## 1 Introduction

Many practical tasks, including medical treatment [50] and content recommendation [31] are commonly modeled within the contextual bandits framework. In the online setting, an agent observes a context at each step and chooses among the available actions. The agent then receives a context-dependent reward corresponding to the action taken, but cannot observe the rewards corresponding to alternative actions. In a healthcare setting, the observed context might be a vector capturing vital signs, lab tests, and other available data, while the action space might consist of the available treatments. The reward to optimize could be a measure of patient health or treatment response.

While contextual bandits research has traditionally focused on the expected reward, stakeholders often care about other risk functionals (parameters of the reward distribution) that express real-world desiderata or have desirable statistical properties. For example, investors assess mutual funds via the Sharpe ratio, which normalizes returns by their variance [44]. Related works in reinforcement learning (RL) have sought to estimate the variance of returns [41, 48] and to optimize the mean return under variance constraints [33]. In safety-critical and financial applications, researchers often measure the conditional value-at-risk (CVaR), which captures the expected return among the lower $\alpha$ quantile of outcomes [40, 28]. In an emerging line of RL works, researchers have explored other risk functionals, including cumulative prospect weighting [22], distortion risk measures [13], and exponential utility functions [16].

35th Conference on Neural Information Processing Systems (NeurIPS 2021).

In many real-world problems otherwise suited to the contextual bandits framework, experimentation turns out to be prohibitively expensive or unethical. In such settings, we might hope to evaluate prospective policies using the data collected under a previous policy. Formally, this problem is called *off-policy evaluation*, and our goal is to evaluate the performance of a target policy $\pi$ using data collected under a behavior policy $\beta$. While most existing research focuses on estimating the expected value of the returns [20, 19], one recent paper evaluates the variance of returns [10].

In this paper, we propose practical methods and the first sample complexity guarantees for *off-policy risk evaluation*, addressing a diverse set of objectives of interest to researchers and practitioners. Towards this end, we introduce *Lipschitz risk functionals* which encompass all objectives for which the risk (i) depends only on the CDF of rewards; and (ii) is Lipschitz with respect to changes in the CDF (as assessed via the sup norm). We prove that for bounded rewards, this class subsumes many risk functionals of practical interest, including variance, mean-variance, conditional value-at-risk, and cumulative prospect weighting, among others.

Thus, given accurate estimates of the CDF of rewards under $\pi$, we can accurately estimate Lipschitz risks. Moreover, (sup norm) bounds on our CDF estimates imply bounds on the corresponding plugin estimates for any Lipschitz risks. The key remaining step is to establish finite sample guarantees on the error in estimating the target policy's CDF of rewards. Our analysis centers on an importance sampling estimator (Section 5.1), and a variance-reduced doubly robust estimator (Section 5.3). We derive finite sample concentrations for both CDF estimators, showing that they achieve the desired $O(1/\sqrt{n})$ rates, where $n$ is the sample size. Moreover, the estimation error for any Lipschitz risk is scales with its Lipschitz constant, and similarly converges as $O(1/\sqrt{n})$.

We assemble these results into an algorithm called OPRA (Algorithm 1) that outputs a comprehensive risk assessment for a target policy $\pi$, using any set of Lipschitz risk functionals. Notably, because all risk estimates share the same underlying CDF estimate, our error guarantees hold simultaneously for all estimated risk functionals in the set, regardless of the cardinality (Section 6). Finally, we present experiments that demonstrate the practical applicability our estimators.

## 2   Related Work

The study of risk functionals and risk-aware algorithms is core to the decision making literature [3, 40, 29, 43, 1, 37, 25]. In the bandit literature, many works address regret minimization problems using risk functionals; popular examples include the CVaR, value-at-risk, and mean-variance [8, 41, 54, 60]. [49] studies optimistic UCB exploration for optimizing CVaR while [11, 6] study Thompson sampling, and [27, 7] study regret minimization for linear combinations of the mean and CVaR. Using the CPT risk functional, [22] considers regret minimization in both $K$-armed bandits and linear contextual bandits. [53, 35] tackle the problem of black-box function optimization under different risk functionals.

In off-policy evaluation, we face an additional challenge due to the discrepancy between the data distribution and that induced by the target policy. Importance sampling (IS) estimators are among the most prominent methods for dealing with distribution shift [2, 23, 45]. Doubly robust (DR) estimators [39, 5] leverage (possibly misspecified) models to achieve lower variance without sacrificing consistency. These estimators have been adapted for off-policy evaluation in multi-armed bandits [32, 52, 10], contextual bandits [20, 19, 57], and Markov decision processes [24, 51]

After deriving our key results, we learned of an independent (also unpublished) work [9] that also employs importance sampling to estimate CDFs for the purpose of providing off-policy estimates for parameters of the reward distribution. However, they do not establish uniform concentration of their CDF estimates or formally relate the errors in CDF and (downstream) parameter estimation leaving open questions concerning the convergence (both in finite samples and asymptotically) of the parameter estimates.

Our work formulates both importance sampling and variance-reduced doubly robust estimators and provides the first uniform finite sample concentration bounds for both types of CDF and risk estimates. For empirical CDF estimation, we build on seminal work by [21], which provided an approximation-theoretic concentration bound that was later tightened by [34].

# 3 Problem Setting

We denote contexts by $X$ and the corresponding context space by $\mathcal{X}$. Similarly, we denote actions by $A$ and the corresponding action space by $\mathcal{A}$. We study the contextual bandit problem characterized by a fixed probability measure over context space $\mathcal{X}$, and a conditional reward distribution $\mathcal{R}(\cdot|X, A)$. In the off-policy setting, we have access to a dataset $\mathcal{D}$ generated using a behavior policy $\beta$ that interacts with the environment for $n$ rounds as follows: at each round, a new context $X$ is drawn and then the policy $\beta$ chooses an action $A \sim \beta(\cdot|X)$. The environment then reveals the reward $R \sim \mathcal{R}(\cdot|X, A)$ for only the chosen action $A$. Running this process for $n$ stepsgenerates a dataset $\mathcal{D} := \{x_i, a_i, r_i\}_{i=1}^n$. In the off-policy evaluation setting, our goal is to evaluate the performance of a target policy $\pi$, using only a dataset $\mathcal{D}$.

Next, we can express our sample space in terms of the contexts, actions, and rewards: $\Omega = (\mathcal{X} \times \mathcal{A} \times \mathbb{R})$. Let $(\Omega, \mathcal{F}, \mathbb{P}_\beta)$ be the probability space induced by the behavior policy $\beta$, and $(\Omega, \mathcal{F}, \mathbb{P})$ the probability space induced by the target policy $\pi$. We assume that $\mathbb{P}$ is absolutely continuous with respect to $\mathbb{P}_\beta$. For any context $x$ and action $a$, the importance weight expresses the ratio between the two densities $w(\omega) = w(a, x) = \frac{\beta(a|x)}{\pi(a|x)}$, and the maximum weight $w_{\max} = \sup_{a,x} w(a, x)$ is simply the supremum taken over all contexts and actions. Further, let $w_2 = \mathbb{E}_{\mathbb{P}_\beta}\left[w(A, X)^2\right]$ denote the exponential of the second order Rényi divergence. Note that by definition, $w_2 \leq w_{\max}$, and in practice, we often have $w_2 \ll w_{\max}$.

Finally, we introduce some notation for describing CDFs: For any $t \in \mathbb{R}$, let $F(t) = \mathbb{E}_\mathbb{P}[\mathbb{1}_{\{R \leq t\}}]$ denote the CDF under the target policy; further, let $G(t; X, A) = \mathbb{E}_\mathbb{P}[\mathbb{1}_{\{R \leq t\}}|X, A] = \mathbb{E}_{\mathbb{P}_\beta}[\mathbb{1}_{\{R \leq t\}}|X, A]$ denote the CDF of rewards conditioned on a context $X$ and action $A$, which is independent of the policy. Lastly, for any $t \in \mathbb{R}$, we denote the variance by $\sigma^2(t; X, A) = \mathbb{V}_\mathbb{P}\left[\mathbb{1}_{\{R \leq t\}}|X, A\right] = \mathbb{V}_{\mathbb{P}_\beta}\left[\mathbb{1}_{\{R \leq t\}}|X, A\right]$.

# 4 Lipschitz Risk Functionals

We now introduce Lipschitz risk functionals, a novel class of objectives for which absolute differences in the risk are bounded by sup norm differences in the CDF of rewards. After formally defining the class, we provide an in-depth review of common risk functionals and their relationship to the CDF of rewards. When possible, we derive the associated Lipschitz constants, when rewards are bounded on support $[0, D]$, relegating all proofs to Appendix A.2.

## 4.1 Defining the Lipschitz Risk Functionals

The Lipschitz risk functionals are a subset of the broader family of *law-invariant risk functionals*. Formally, let $Z \in \mathcal{L}_\infty(\Omega, \mathcal{F}_Z, \mathbb{P}_Z)$ denote a real-valued random variable that admits a CDF $F_Z \in \mathcal{L}_\infty(\mathbb{R}, \mathbb{B}(\mathbb{R}))$. A *risk functional* $\rho$ is a mapping from a space of random variables to the space of real numbers $\rho : \mathcal{L}_\infty(\Omega, \mathcal{F}_Z, \mathbb{P}_Z) \to \mathbb{R}$. Any risk functional $\rho$ is said to be law-invariant if $\rho(Z)$ depends only on the distribution of $Z$ [30].

**Definition 4.1** (Law-Invariant Risk Functional). *A risk functional $\rho : \mathcal{L}_\infty(\Omega, \mathcal{F}, \mathbb{P}) \to \mathbb{R}$, is law-invariant if for any pair of random variables $Z$ and $Z'$, $F_Z = F_{Z'} \implies \rho(Z) = \rho(Z')$.*

When clear from the context, we sometimes abuse notation by writing $\rho(F_Z)$ in place of $\rho(Z)$. In general, it may not be practical to estimate risk functionals that are not law invariant from data [4]. Thus focusing on law-invariant risks is only mildly restrictive.

We can now formally define the Lipschitz risk functionals:

**Definition 4.2** (Lipschitz Risk Functional). *A law invariant risk functional $\rho$ is $L$-Lipschitz if for any pair of CDFs $F_Z$ and $F_{Z'}$ and some $L \in (0, \infty)$, it satisfies*

$$|\rho(F_Z) - \rho(F_{Z'})| \leq L\|F_Z - F_{Z'}\|_\infty.$$

A risk functional is $L$-Lipschitz if, for any two random variables $Z, Z'$, its value is upper bounded by the sup-norm of the difference between their corresponding CDFs. The significance of this Lipschitzness property in the contextual bandit setting is that, given a high confidence bound on the error of the estimated CDF of rewards for a policy $\pi$, we can obtain a high confidence bound on its evaluation under any $L$-Lipschitz law-invariant risk functional on the distribution of rewards.

## 4.2 Overview of Common Risk Functionals (and their Lipschitzness)

We now briefly describe some popular classes of risk functionals and, when possible, derive their associated Lipschitz constants.

**Coherent Risk Functionals**   The set of risk functionals that satisfy properties called monotonicity, subadditivity, translation invariance, and positive homogeneity (see Appendix A.2), constitute the *coherent risk functionals* [3, 15]. While not all coherent risk functionals are law-invariant, nearly among those commonly addressed in the literature are. Examples include expected value, conditional value-at-risk (CVaR), entropic value-at-risk, and mean semideviation [11, 49, 47, 43]. Others include the Wang transform function [55] and the proportional hazard (PH) risk functional [58].

**Distorted Risk Functionals**   When the random variable $Z$ is required to be non-negative, law-invariant coherent risk functionals are examples of the more general class of law-invariant *distorted risk functionals* [17, 55, 56, 4]. For $Z \geq 0$, a distorted risk functional has the following form

$$C,$$

where the distortion function $g : [0,1] \to [0,1]$ is an increasing function with $g(0) = 0$ and $g(1) = 1$. Distorted risk functionals are coherent if and only if $g$ is concave [58]. For example, when $g(s) = \min\{\frac{s}{1-\alpha}, 1\}$ for $s \in [0,1]$ and $\alpha \in (0,1)$, CVaR at level $\alpha$ is recovered. When $g$ is the identity map, the distorted risk functional is the expected value. The Wang risk functional at level $\alpha$ [55] is recovered when $g(s) = F(F^{-1}(s) - F^{-1}(\alpha))$, and the proportional hazard risk functional can by obtained by setting $g(s) = s^\alpha$ for $\alpha < 1$. Not all distorted risk functionals are coherent. For example, setting $g(s) = \mathbb{1}_{\{s \geq 1-\alpha\}}$ recovers the value-at-risk (VaR), which is not coherent.

**Lemma 4.1** (Lipschitzness of Coherent and Distorted Risk Functionals). *On the space of random variables with support in $[0, D]$, the distorted risk functional of any $\frac{L}{D}$-Lipschitz distortion function $g : [0,1] \to [0,1]$, i.e., $|g(t) - g(t')| \leq \frac{L}{D}|t - t'|$, is a $L$-Lipschitz risk functional.*

**Remark 4.1** (Expected Value and CVaR). *Both expected value and CVaR are examples of distorted risk functionals. Then using Lemma 4.1, on the space of random variables with support in $[0, D]$, the expected value risk functional is $D$-Lipschitz because $g$ is the identity and thus 1-Lipschitz. On the same space, the risk functional $CVaR_\alpha$ is $\frac{D}{\alpha}$-Lipschitz because $g$ is $\frac{1}{\alpha}$-Lipschitz.*

**Cumulative Prospect Theory (CPT) Risk Functionals**   CPT risks [37] take the form:

$$\rho(F_Z) = \int_0^{+\infty} g^+ \left(1 - F_{u^+(Z)}(t)\right) dt - \int_0^{+\infty} g^- \left(1 - F_{u^-(Z)}(t)\right) dt,$$

where $g^+, g^- : [0,1] \to [0,1]$ are continuous functions with $g^{+/-}(0) = 0$ and $g^{+/-}(1) = 1$. The functions $u^+, u^- : \mathbb{R} \to \mathbb{R}_+$ are continuous, with $u^+(z) = 0$ when $z \geq c$ and $u^-(z) = 0$ when $z < c$ for some constant $c \in \mathbb{R}$. Importantly, the CPT functional handles gains and losses separately. The functions $u^+, u^-$ compare the random variable $Z$ to a baseline $c$, and the distortion $g^+$ is applied to "gains" (when $Z \geq c$), while $g^-$ is applied to "losses" (when $Z < c$).

**Lemma 4.2** (Lipschitzness of CPT Functional). *On the space of random variables with support in $[0, D]$, if the CPT distortion functions $g^+$ and $g^-$ are both $\frac{L}{D}$-Lipschitz, then the CPT risk functionals is $L$-Lipschitz.*

**Other Risk Functionals**   The variance, mean-variance, and many other popular risks do not fit easily into the aforementioned classes, but are nevertheless law-invariant. For example, for a nonnegative random variable $Z$, the variance is defined as $\rho(F_Z) = 2 \int_0^\infty t(1 - F_Z(t))dt - \left(\int_0^\infty (1 - F_Z(t))dt\right)^2$. Moreover, the variance and mean-variance are both $L$-Lipschitz.

**Lemma 4.3** (Lipschitzness of Variance). *On the space of random variables with support in $[0, D]$, variance is a $3D^2$-Lipschitz risk functional.*

A number of recent papers have addressed risk functionals expressed as weighted combinations of others, e.g., mean-variance [41]. Other papers have optimized constrained objectives, such as expected reward constrained by variance or CVaR below a certain threshold [12, 38]. When expressed as Lagrangians, these objectives can also be expressed as weighted combinations of the risk functionals involved. We extend the Lipschitzness property to risk functionals of this form:

**Lemma 4.4** (Lipschitzness of Weighted Sum of Risk Functionals). *Let $\rho$ be a weighted sum of risk functionals $\rho_1, ..., \rho_K$ that are $L_1, ..., L_K$-Lipschitz, respectively, with weights $\lambda_1, ...., \lambda_K > 0$, i.e., $\rho(Z) = \sum_{k=1}^{K} \lambda_k \rho_k(Z)$. Then $\rho$ is $\sum_k \lambda_k L_k$-Lipschitz.*

**Remark 4.2.** *Note that mean-variance is given by $\rho(Z) = \mathbb{E}[Z] + \lambda \mathbb{V}(Z)$ for some $\lambda > 0$. Then, using Lemma 4.4, we immediately obtain that mean-variance is $(1 + 3\lambda D^2)$-Lipschitz.*

Though we have provided many examples of Lipschitz risk functionals in this section, it is worth noting that there are a number of risk functionals that do not satisfy the Lipschitzness property, such as the value-at-risk (VaR). For the sake of brevity, we omit consideration of such risk functionals in this paper, and outline future avenues of research on this topic in the discussion.

## 5 Off-Policy CDF Estimation

This section describes our method for high-confidence off-policy estimation of $F$, the CDF of returns under the policy $\pi$. The key challenge in estimating $F$ is that the reward samples are observed only for actions taken by the behavior policy $\beta$. To overcome this limitation, one intuitive solution is to reweight the observed samples according to their importance sampling (IS) weight (Section 5.1). However, IS estimators are known to suffer from high variance. To mitigate this, we define the first doubly robust CDF estimator (Section 5.3).

### 5.1 CDF Estimation with Importance Sampling (IS)

Given an off-policy dataset $\mathcal{D} = \{x_i, a_i, r_i\}_{i=1}^{n}$, we define the following nonparametric IS-based estimator for the empirical CDF,

$$\widehat{F}_{\text{IS}}(t) := \frac{1}{n} \sum_{i=1}^{n} w(a_i, x_i) \mathbb{1}_{\{r_i \le t\}}, \tag{1}$$

where $w(a, x) = \frac{\pi(a|x)}{\beta(a|x)}$ are the importance weights. The IS estimator is pointwise-unbiased, with variance given below (proof in Appendix B.1):

**Lemma 5.1.** *The IS estimator* (1) *is unbiased and its variance is*

$$\mathbb{V}_{\mathbb{E}_\beta}\left[\widehat{F}_{IS}(t)\right] = \frac{1}{n} \mathbb{E}_{\mathbb{P}_\beta}\left[w(A, X)^2 \sigma^2(t; X, A)\right] + \frac{1}{n} \mathbb{V}_{\mathbb{P}_\beta}\left[\mathbb{E}_{\mathbb{P}_\beta}\left[w(A, X)G(t; X, A)|X\right]\right]$$
$$+ \frac{1}{n} \mathbb{E}_{\mathbb{P}_\beta}\left[\mathbb{V}_{\mathbb{P}_\beta}\left[w(A, X)G(t; X, A)|X\right]\right].$$

The expression for variance is broken down into three terms. The first term represents randomness in the rewards. The second term represents variance due to the randomness over contexts $X$. The final term is the penalty arising from using importance sampling, and is proportional to the importance sampling weights $w$ and the true CDF of conditional rewards $G$. The variance contributed by the third term can be large when the weights $w$ have a wide range, which occurs when $\beta$ assigns extremely small probabilities to actions where $\pi$ assigns high probability.

Due to the use of importance sampling weights, the estimated CDF $\widehat{F}_{\text{IS}}(t)$ may be greater than 1 for some $t$, even though a valid CDF must be in the interval $[0, 1]$ for all $t$. To mitigate this problem, a weighted importance sampling (WIS) estimator can be used, which normalizes each importance weight by the sum of importance weights:

$$\widehat{F}_{\text{WIS}}(t) = \frac{1}{\sum_{j=1}^{n} w(a_j, x_j)} \sum_{i=1}^{n} w(a_i, x_i) \mathbb{1}_{\{r_i \le t\}},$$

which [9] shows is a biased but uniformly consistent estimator. Another option is the clipped estimator IS-Clip (2), which simply limits the estimator to the unit interval:

$$\widehat{F}_{\text{IS-CLIP}}(t) := \min\{\widehat{F}_{\text{IS}}(t), 1\}. \tag{2}$$

Although $\widehat{F}_{\text{IS-CLIP}}$ has lower variance than the IS estimator, it is potentially biased. However, given finite samples, we can bound with high confidence the sup-norm error between $\widehat{F}_{\text{IS-CLIP}}$ and $F$, in Theorem 5.1 below (proof in Appendix B.2):

**Theorem 5.1.** *Given $n$ samples drawn from $\mathbb{P}_\beta$, for the IS estimator $\widehat{F}_{\mathrm{IS}}$, we have*

$$\mathbb{P}_\beta\left(\|\widehat{F}_{\mathrm{IS}} - F\|_\infty \leq \varepsilon_{\mathrm{IS}_1} := \sqrt{\frac{8w_{\max}^2}{n}\log(4/\delta)}\right) \geq 1 - \delta. \tag{3}$$

*or, based on $w_2$, we obtain a Bernstein-style bound,*

$$\mathbb{P}_\beta\left(\left\|\widehat{F}_{\mathrm{IS}} - F\right\|_\infty \leq \varepsilon_{IS_2} := \frac{4w_{\max}\log(4/\delta)}{n} + 2\sqrt{\frac{2w_2\log(4/\delta)}{n}}\right) \geq 1 - \delta. \tag{4}$$

*The same bounds hold for $\|\widehat{F}_{\mathrm{IS\text{-}CLIP}} - F\|_{\mathcal{L}_\infty}$.*

When $w_2 \ll w_{max}$, we observe that inequality (4) is more favorable than inequality (3). Theorem 5.1 demonstrates that the $\widehat{F}_{\mathrm{IS\text{-}CLIP}}$ uniformly converges to the true CDF at a rate of $O(1/\sqrt{n})$, with the uniform consistency of $\widehat{F}_{\mathrm{IS\text{-}CLIP}}$ as an immediate consequence. To the best of our knowledge, it is the first DKW-style concentration inequality on the importance sampling estimator CDF estimator in off-policy evaluation. The bound has explicit constants and subsumes the classical DKW inequality.

## 5.2 Model-Based CDF Estimation

As we have shown previously, IS estimators can suffer from high variance, which can be limiting in practice. However, in many practical applications, we may have access to a model $\overline{G}(t; X, A)$ of the conditional distribution $G(t; X, A)$, which can be used in estimation with very low variance. In many cases, practitioners may have a model of $\overline{G}$ from expert studies or from a simulator, or can form a regression estimate of $\overline{G}$ from logged data. One simple model-based estimator can then be obtained using the *direct method*, which simply employs the model $\overline{G}$ for each observed context:

$$\widehat{F}_{\mathrm{DM}}(t) = \frac{1}{n}\sum_{i=1}^n \overline{G}(t; x_i, \pi), \quad \text{where } \overline{G}(t; x_i, \pi) = \sum_a \pi(a|x_i)\overline{G}(t; x_i, a). \tag{5}$$

Because the DM estimator $\widehat{F}_{\mathrm{DM}}$ does not use importance weights, it can have significantly lower variance than the IS and DR estimators. In general, however, the DI estimator is biased, and its error flows directly from error in the model $\overline{G}$ (full derivations of the bias and variance are given in Lemma E.3 of Appendix E). The magnitude and distribution of bias over the context and action space is difficult to characterize. In practice, $\overline{G}$ is often estimated or modeled agnostic to the target policy, and hence may not be well-approximated in areas that are important for $\pi$. If $\overline{G}$ is an accurate model of the conditional reward distribution, however, then $\widehat{F}_{\mathrm{DM}}$ is a good approximation of $F$.

## 5.3 Doubly Robust (DR) CDF Estimation

We now define a doubly robust (DR) CDF estimator that takes advantage of both importance sampling and models $\overline{G}$ to obtain the best characteristics of both types of estimation. In particular, the DR estimator is unbiased, but has potentially significant reduction in variance. The DR estimator for the empirical CDF is defined to be

$$\widehat{F}_{\mathrm{DR}}(t) := \frac{1}{n}\sum_{i=1}^n w(a_i, x_i)\left(\mathbb{1}_{\{r_i \leq t\}} - \overline{G}(t; x_i, a_i)\right) + \overline{G}(t; x_i, \pi), \tag{6}$$

where $\overline{G}(t; x, \pi) = \mathbb{E}_{\mathbb{P}_\beta}\left[\overline{G}(t; x, A)|x\right]$. Informally, the DR estimator takes the model $\overline{G}$ as a baseline, using the available data to apply a correction. While $\overline{G}$ alone may be biased, $\widehat{F}_{\mathrm{DR}}$ is an unbiased estimator of $F$, and can have reduced variance compared to the IS estimator (proof in Appendix C.1):

**Lemma 5.2.** *The DR estimator* (6) *is unbiased and its variance is*

$$\mathbb{V}_{\mathbb{P}_\beta}\left[\widehat{F}_{\mathrm{DR}}(t)\right] = \frac{1}{n}\mathbb{E}_{\mathbb{P}_\beta}\left[w(A, X)^2\sigma^2(t; X, A)\right] + \frac{1}{n}\mathbb{V}_{\mathbb{P}_\beta}\left[\mathbb{E}_{\mathbb{P}_\beta}\left[w(A, X)G(t; X, a)|X\right]\right]$$

$$+ \frac{1}{n}\mathbb{E}_{\mathbb{P}_\beta}\left[\mathbb{V}_{\mathbb{P}_\beta}\left[w(A, X)\left(G(t; X, A) - \overline{G}(t; X, A)\right)|X\right]\right].$$

The variance reduction advantage of the DR estimator becomes apparent from a direct comparison of the three terms in the IS estimator variance (Lemma 5.1) and the DR estimator variance (Lemma 5.2). The first and second terms, which capture the variance in rewards and contexts, are identical. The third term, which represents the importance sampling penalty, is proportional to $G - \overline{G}$ in the DR estimator, but proportional to $G$ in the IS estimator. When this difference $G - \overline{G}$ is smaller than $G$, which is often the case in practice, the third term has reduced variance in the DR estimator. The magnitude of variance reduction is greater when the weights $w$ have a large range, which is precisely when large variance can become problematic in importance sampling.

**Remark 5.1** (Double Robustness). *Although we consider the setting where the behavior policy $\beta$ is known, when $\beta$ is unknown and needs to be estimated, the estimator $\widehat{F}_{\text{DR}}$ is consistent when either $\overline{G}$ is consistent or the policy estimator is consistent. This is where the name "doubly robust" comes from. We demonstrate and discuss this fact further in Appendix E.*

Although the DR estimator $\widehat{F}_{\text{DR}}$ has desirable reductions in variance, given finite samples, it is not guaranteed to be a valid CDF. Like the IS estimator, the DR estimator may be greater than 1 for some $t$ due to the use of importance weighting. However, it may also be negative at some $t$ as a consequence of the subtracted term in (6). As an additional consequence of this term, the DR estimator is not guaranteed to be a monotone function. As a result, in order to use the DR CDF estimate for risk estimation, we must transform $\widehat{F}_{\text{DR}}$ into a monotone function bounded in $[0, 1]$. Examples of such transformations include isotonic approximation [46] and monotone $\mathcal{L}_p$ approximation [14].

For our analysis, however, we consider a simple monotone transformation that involves an accumulation function, which does not allow the CDF to decrease, followed by a clipping to $[0, 1]$:

$$\widehat{F}_{\text{M-DR}}(t) = \text{Clip}\left\{\max_{t' \leq t} \widehat{F}_{\text{DR}}(t'), 0, 1\right\}, \tag{7}$$

which is a uniformly consistent estimator, as the following concentration guarantee shows:

**Theorem 5.2.** *The monotone transformation of the DR estimator $\widehat{F}_{\text{M-DR}}(t)$ satisfies*

$$\mathbb{P}_\beta\left(\left\|\widehat{F}_{\text{M-DR}} - F\right\|_\infty \leq \epsilon_{\text{DR}} := \sqrt{\frac{72 w_{\max}^2}{n} \log\left(\frac{8 n^{1/2}}{\delta}\right)}\right) \geq 1 - \delta. \tag{8}$$

The proof is in Appendix C.2. The purpose of Theorem 5.2 is to show the dependence of the error on the importance weights $w_{max}$ and on the finite sample size $n$. Using the M-DR estimator, we again recover a sample complexity of $\widetilde{O}\left(1/\sqrt{n}\right)$. The proof is given in Appendix C.2. Note that (8) does not depend on the error $\overline{G} - G$, which is the term responsible for variance reduction, as given in Lemma E.5. A tighter bound that incorporates the error $G - \overline{G}$ remains an open problem.

# 6 Off-Policy Risk Assessment

Given any law-invariant risk functional $\rho$ and CDF estimator $\widehat{F}$, we can estimate the value of the risk functional as $\widehat{\rho} := \rho(\widehat{F})$. However, the estimator $\widehat{\rho}$ may be biased even if $\widehat{F}$ is unbiased. For Lipschitz risk functionals introduced in Section 4, we can obtain their finite sample error bounds, using the error bound of the CDF estimator. Further, a set of risk functionals of interest can be evaluated using the same estimated CDF, which suggests that the error bound of the CDF gives error bounds on the risk estimators that hold simultaneously.

Theorem 6.1 utilizes our error bound of the estimated CDF to derive error bounds for estimators of a set of Lipschitz risk functionals. As we showed in Section 4, most if not all commonly studied risk functionals satisfy the property of Lipschitzness, showing our result's wide applicability.

**Theorem 6.1.** *Given a set of Lipschitz risk functionals $\{\rho_p\}_{p=1}^{P}$ with Lipschitz constants $\{L_p\}_{p=1}^{P}$, and a CDF estimator $\widehat{F}$, such that $\|\widehat{F} - F\|_\infty \leq \epsilon$ with probability at least $1 - \delta$, we have with probability at least $1 - \delta$ that for all $p \in \{1, \ldots P\}$,*

$$\left|\rho_p(\widehat{F}) - \rho_p(F)\right| \leq L_p \epsilon.$$

Thus, one powerful property of risk estimation using the estimated CDF approach is that, given a high-probability error bound on the CDF estimator, the corresponding error bounds on estimates of *all* Lipschitz risk functionals of interest hold *simultaneously* with the same probability. Further, because the error of the IS CDF estimator $\epsilon_{\text{IS}}$ (Theorem 5.1) and DR CDF estimator $\epsilon_{\text{DR}}$ (Theorem 5.2) converge at a rate of $O(1/\sqrt{n})$, Theorem 6.1 shows that the error of all Lipschitz risk functional estimators shrink at a rate of $O(1/\sqrt{n})$. Thus, $\rho_p(\widehat{F})$ are consistent risk functional estimators.

**Remark 6.1.** *When $\rho$ is the expected return, the Lipschitz constant is $L = D$, where $D$ is the maximum reward, and we achieve a convergence rate of $\mathcal{O}(Dw_{max}/\sqrt{n})$. This matches the rate of error bounds for off-policy mean estimation previously derived in [20].*

Putting these results together, we now provide an algorithm, called OPRA (Algorithm 1), which given an off-policy contextual bandit dataset and a set of Lipschitz risk functionals of interest, outputs for each risk functional an estimate of its value and a confidence bound. The algorithm first uses a valid CDF estimator, e.g., the clipped IS estimator (2) or monotonized DR estimator (7), to form $\widehat{F}$ with sup-norm error $\epsilon$. OPRA then evaluates each $L_p$-Lipschitz risk functional $\rho_p$ on $\widehat{F}$ to obtain $\widehat{\rho}_p$, along with its upper and lower confidence bound $\widehat{\rho}_p \pm L_p\epsilon$.

---

**Algorithm 1:** Off-Policy Risk Assessment (OPRA)

---

**Input:** Dataset $\mathcal{D}$, policy $\pi$, probability $\delta$, models $\overline{G}$, Lipschitz risk functionals $\{\rho_p\}_{p=1}^P$ with Lipschitz constants $\{L_p\}_{p=1}^P$.

1 Estimate the CDF using a valid CDF estimator $\widehat{F}$;
2 Compute the corresponding CDF estimation error $\epsilon$ such that $\mathbb{P}(\|F - \widehat{F}\|_\infty < \epsilon) \geq 1 - \delta$;
3 **for** *p = 1 ... P* **do**
4 $\quad$ Estimate $\widehat{\rho}_p = \rho_p(\widehat{F})$;
5 **end**
**Output:** Estimates with errors $\{\widehat{\rho}_p \pm L_p\epsilon\}_{p=1}^P$.

---

OPRA can be used to obtain a full risk assessment of any given policy, using the input Lipschitz risk functionals of interest, which can include the popularly used mean, variance, and CVaR. As demonstrated in Theorem 6.1, the error guarantee on the risk estimators holds simultaneously for all $P$ risk functionals with probability at least $1 - \delta$. Importantly, OPRA also demonstrates the computational efficiency of the distribution-centric risk estimation approach proposed in this paper. For a given $\pi$, the CDF only needs to be estimated once, and can be used repetitively to estimate the value of the risk functionals. Further, the error of the risk estimators are determined by the known error of the CDF estimator, multiplied by the known Lipschitz constants.

## 7 Empirical Studies

In this section, we give empirical evidence for the effectiveness of the doubly robust (DR) CDF and risk estimates, in comparison to the importance sampling (IS), weighted importance sampling (WIS), and direct method (DM) estimates. Further, we demonstrate the convergence of the CDF and risk estimation error in terms of the number of samples. In our experiments, the relevant baselines are the IS and WIS CDF estimators with plugin risk estimates, both of which were contemporaneously proposed by [9], which are the only existing off-policy estimators for general risks. In most of our experiments, the DR CDF and risk estimates exhibit reduced variance compared to the baselines, a possibility suggested by our theoretical results.

### 7.1 UCI Datasets

**Setup.** Following [20, 19, 57], we obtain our off-policy contextual bandit datasets by transforming classification datasets. The contexts are the provided features, and the actions correspond to the possible class labels. To obtain the evaluation policy $\pi$, we use the output probabilities of a trained logistic regression classifier. The behavior policy is defined as $\beta = \alpha\pi + (1 - \alpha)\pi_{\text{UNIF}}$, where $\pi_{\text{UNIF}}$ is a uniform policy over the actions, for some $\alpha \in (0, 1]$. We apply this process to the PageBlocks and OptDigits datasets [18], which have dimensions $d$ and actions $k$ using $\alpha = 0.1$ (Figure 1). When models $\overline{G}$ are used (for DM, DR estimators), as in [20], the dataset is divided into two splits, with

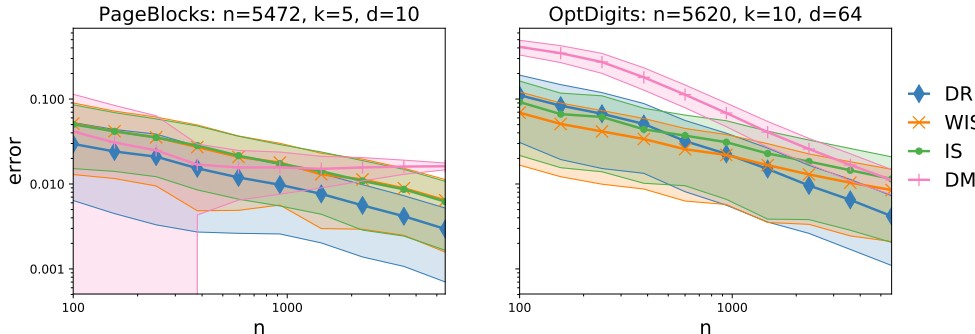

Figure 1: The error of the CDF estimators as a function of sample size $n$, for **(left)** the PageBlocks dataset and **(right)** the OptDigits dataset. Shaded area is the 95% quantile over 500 runs.

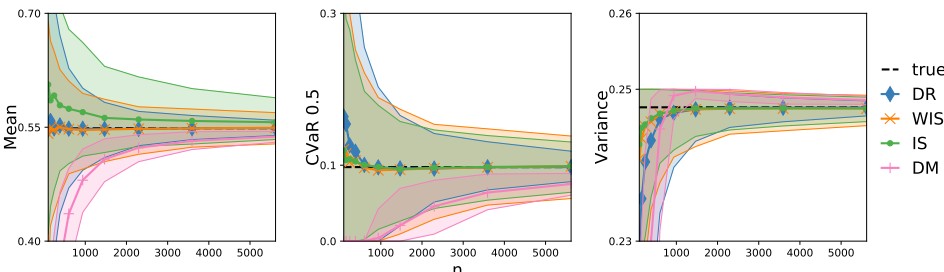

Figure 2: Estimated mean, $\text{CVaR}_{0.5}$, and variance for the OptDigits dataset, compared to their true values (black). Shaded area is the standard deviation over 500 runs.

each of the two splits used to calculate $\overline{G}$ via regression, which is then used with the other split to calculate the estimator. The two results are averaged to produce the final estimators. We provide further details and extensive evaluations in Appendix F.

**CDF Estimation.** We evaluate the error $\|F - \widehat{F}\|_\infty$ of our CDF estimators against sample size for two UCI datasets (Figure 1). The IS and DR exhibit the expected $O\left(1/\sqrt{n}\right)$ rate of convergence in error previously derived in Theorems 5.1 and 5.2, respectively. We note that the WIS estimator, while biased, performs as well as the IS estimator if not better. In the PageBlocks dataset (Figure 1, left), the regression model for $\overline{G}$ is relatively well-specified as exemplified by the relatively low error of the DM estimator, though it has high variance for low samples sizes. The DR estimator leverages this model to outperform all other estimators for all sample sizes, without suffering the drawbacks of the DM estimator. It takes an order of magnitude less data to reach the same error compared to the IS and WIS estimators. In contrast, the regression model is less well-specified in the OptDigits dataset for lower sample sizes (Figure 1, right), and consequently, the DR estimator cannot perform as well as the IS and WIS estimators for small $n$. However, this trend reverses as data increases and the model improves, with the DR estimator outperforming the importance sampling estimators.

**Risk Functional Estimation.** Figure 2 shows the mean, variance, and $\text{CVaR}_{0.5}$ estimates, which are obtained by evaluating each risk functional on the CDF estimators for the OptDigits dataset. Here, the estimates are plotted against the true value (dashed line) to make the variance reduction effect of the DR estimators more apparent. The DM estimator, which appeared to have competitive performance in the CDF error plot, has relatively high risk estimate error, which occurs because the DM CDF may be poorly approximated in areas that are important for risk functional estimation. The IS, WIS, and DR risk estimates converge quickly to the true value as $n$ increases, and as expected, their relative behavior echoes the trends in Figure 1 as a consequence of our distributional approach. The DR estimator has slightly worse performance for small samples sizes due to the poor specification of the model, but soon exhibits the desired variance reduction for $n > 1000$.

**Comparison to Other Risk Estimators.** One natural question to ask is how OPRA risk estimates, which can be generated for many risk functionals simultaneously, compare to existing estimators for

individual risk functionals. To the best of our knowledge, of the risk functionals described in this paper, previous work has only derived off-policy estimators for the mean and variance.

[20] establishes IS and DR estimators for the mean, and [10] proposes an IS estimator for the variance. We compare these risk estimates to OPRA in Table 1, using the Pageblocks dataset of Figure 1 at select sample sizes $n$. We note that while a number of enhancements have been proposed for off-policy mean estimator, e.g., clipping importance weights [57], we compare only to analogous estimators. In general, the OPRA DR estimates outperform all other estimators. The OPRA DR estimate of the mean have the same error as the DR mean estimator from [20], and OPRA IS estimates of both mean and variance have lower MSE than existing IS mean and variance estimators.

| | Mean | | | | Variance | | | |
|---|---|---|---|---|---|---|---|---|
| | IS [20] | OPRA IS | DR [20] | OPRA DR | IS [10] | OPRA IS | DR | OPRA DR |
| $n = 100$ | 7.5e-3 | 4.4e-3 | 1.6e-3 | **1.6e-3** | 5.0e-5 | 0.7e-5 | N/A | **0.6e-5** |
| $n = 924$ | 7.5e-4 | 4.7e-4 | 2.3e-4 | **2.3e-4** | 4.0e-6 | 0.7e-6 | N/A | **0.7e-6** |
| $n = 5472$ | 12.5e-5 | 7.1e-5 | 3.4e-5 | **3.4e-5** | 5.4e-7 | **0.9e-7** | N/A | 1.1e-7 |

Table 1: Comparison of mean squared error (MSE) for OPRA versus existing off-policy estimators of the mean and variance on the Pageblocks dataset (averaged over 1000 repetitions).

## 7.2 Application: Diabetes Treatment

To demonstrate the efficacy of our method in more complex applications, we use OPRA to evaluate the risks of a target policy for diabetes treatment in the Simglucose simulator [59]. In Simglucose, the agent must control insulin bolus injections given to a type 1 diabetes patient over the course of a day, and the state is a continuous vector consisting of the patient's carbohydrate intake from the last meal, blood glucose levels, and other patient measurements. At the end of the day, the agent receives a reward proportional to how often the patient's blood glucose levels stayed within acceptable limits.

We define the target policy $\pi$ to be the simulator's built-in controller for the patient, and estimate the CDF, mean reward, and $\text{CVaR}_{0.05}$ of $\pi$. The DR estimator has lower error for CDF and risk estimation compared to all baseline estimators, reinforcing our findings on the UCI data (Figure 3).

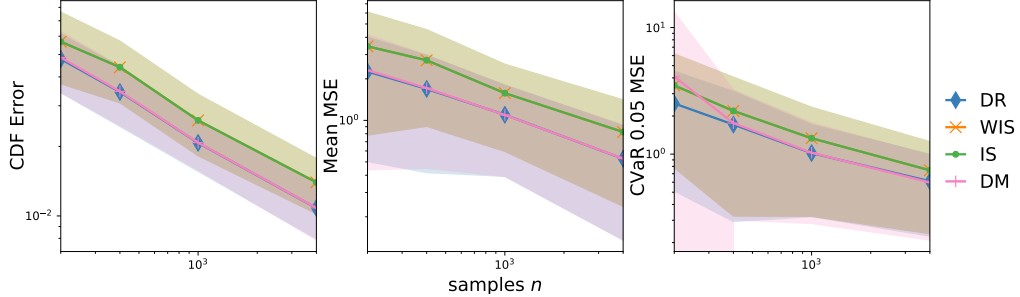

Figure 3: CDF and risk estimation error of OPRA in Simglucose (averaged over 100 repetitions).

## 8 Conclusion

This paper introduces a CDF-centric method for high confidence off-policy estimation of risk functionals, and the first doubly robust CDF estimator. From a theoretical point of view, we provide the first finite-sample concentration inequalities and confidence intervals for a variety of CDF and risk estimators, which are widely applicable in distributional RL settings [13, 28]. From a practical standpoint, our method can be used to comprehensively evaluate the behavior of a target policy before deployment under a wide range of risk functionals. Our work suggests several directions for future work. First, our uniform concentration bound for the doubly robust CDF estimator might be improved by incorporating its variance. Second, our heuristics for transforming CDF estimators into valid cumulative distribution functions, could likely be improved. Finally, our work suggests a natural question: can concentration inequalities for risk functional estimators be derived if and only if the risk functional is Lipschitz, in some general sense?

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
