# Contents (Appendix)

# A  Appendix for Risk Functionals (Section 4)

## A.1  Review of Risk Functionals

In Section 4.1 we introduced several classes of risk functionals and popular examples of each class. We now provide a formal axiomatic definition for classes of risk functionals, and begin by enumerating a set of prominent axioms used to describe risk functionals in the current literature [3, 42]. Classes of risk functionals are defined by the axioms that they satisfy, and one can define a class by choosing the subset best suited to the problem at hand.

**Definition A.1** (Axioms of Risk Functionals). *Consider a pair of random variables $Z$ and $Z'$ and a risk functional $\rho$. We have the following axioms:*

1. *Monotonicity: $\rho(Z) \leq \rho(Z')$ whenever $Z \leq Z'$.*

2. *Subadditivity: $\rho(Z + Z') \leq \rho(Z) + \rho(Z')$.*

3. *Additivity: $\rho(Z + Z') = \rho(Z) + \rho(Z')$ if $Z$ and $Z'$ are co-monotonic random variables (i.e., there exists a random variable $Y$ and weakly increasing functions $f, g$ such that $Z = f(Y)$ and $Z' = g(Y)$).*

4. *Translation invariance: $\rho(Z + c) = \rho(Z) + c, \forall c \in \mathbb{R}$.*

5. *Positive homogeneity: $\rho(tZ) = t\rho(Z)$ for $t > 0$.*

6. *Bounded above by the maximum cost, i.e., $\rho(Z) \leq \max(Z)$.*

7. *Bounded below by the mean cost, i.e., $\rho(Z) \geq \mathbb{E}[Z]$.*

**Coherent Risk Functionals.**  The set of risk functionals that satisfy monotonicity (Axiom 1), subadditivity (Axiom 2), translation invariance (Axiom 4), and positive homogeneity (Axiom 5), constitute the class of *coherent risk functionals* [3, 15]. Further, if a law-invariant coherent risk functional additionally satisfies Additivity (Axiom 3), it is said to be a *spectral risk functional* [26, 1].

**Distorted Risk Functionals.**  Distorted risk functionals have many desirable theoretical properties. They are translation invariant (Axiom 4) and positive homogeneous (Axiom 5), and are defined utilizing (Axiom 6) and (Axiom 7) [58]. They satisfy Axiom 7 if and only if $g(s) \geq s \ \forall s \in [0, 1]$ [58], and are subadditive (Axiom 2) if and only if $g$ is concave, which preserves second order stochastic dominance [55]. In addition, all distorted risk functionals preserve stochastic first order dominance [58].

**CPT-Inspired Risk Functionals.**  In general, due to the separate consideration of losses and gains, the CPT-inspired risk functional may not satisfy any of the above axioms. However, additional assumptions on the distortions $g^+$ and $g^-$ may allow certain axioms to be satisfied. For example, if the random variable has nonnegative support and the threshold $c$ is set to be 0 so that only gains are observed, and $g^+$ is additionally increasing, we recover the distorted risk functionals with axioms specified above. If $g^+$ is additionally concave, then we recover the coherent risk functionals.

## A.2  Proofs for Risk Functionals

*Proof of Lemma 4.1.*

$$\left| \rho\left(F_Z\right) - \rho\left(F_{Z'}\right) \right| = \left| \int_0^D g\left(1 - F_Z(t)\right) - g\left(1 - F_{Z'}(t)\right) dt \right|$$

$$\leq \int_0^D \left| g\left(1 - F_Z(t)\right) - g\left(1 - F_{Z'}(t)\right) \right| dt$$

$$\leq \int_0^D \frac{L}{D} \left| F_{Z'}(t) - F_Z(t) \right| dt$$

$$\leq L \max_t \left| F_Z(t) - F_{Z'}(t) \right|,$$

where the second to last step uses the $L/D$-Lipschitzness of $\rho$. $\qquad\square$

*Proof of Lemma 4.2.* Using the definition of the CDF, note that on the bounded support of $[0, D]$ the CPT functional can be rewritten as

$$\rho(F_Z) = \int_0^D g^+ \left(\mathbb{P}_Z \left(u^+(Z) > t\right)\right) dt - \int_0^D g^- \left(\mathbb{P}_Z \left(u^-(Z) > t\right)\right) dt.$$

Then,

$$
\begin{aligned}
|\rho(Z) - \rho(Z')| &= \left| \int_0^D g^+ \left(\mathbb{P}_Z \left(u^+(Z) > t\right)\right) dt - \int_0^D g^- \left(\mathbb{P}_Z \left(u^-(Z) > t\right)\right) dt \right. \\
&\qquad \left. - \int_0^D g^+ \left(\mathbb{P}_{Z'} \left(u^+(Z') > t\right)\right) dt - \int_0^D g^- \left(\mathbb{P}_{Z'} \left(u^-(Z') > t\right)\right) dt \right| \\
&\leq \left| \int_0^D g^+ \left(\mathbb{P}_Z \left(u^+(Z) > t\right)\right) dt - \int_0^D g^+ \left(\mathbb{P}_{Z'} \left(u^+(Z') > t\right)\right) dt \right| \\
&\qquad + \left| \int_0^D g^- \left(\mathbb{P}_Z \left(u^-(Z) > t\right)\right) dt - \int_0^D g^- \left(\mathbb{P}_{Z'} \left(u^-(Z') > t\right)\right) dt \right| \\
&\leq \frac{L}{D} \int_0^D \left| \mathbb{P}_Z \left(u^+(Z) > t\right) - \mathbb{P}_{Z'} \left(u^+(Z') > t\right) \right| dt \\
&\qquad + \frac{L}{D} \int_0^D \left| \mathbb{P}_Z \left(u^-(Z) > t\right) - \mathbb{P}_{Z'} \left(u^-(Z') > t\right) \right| dt \\
&\leq \frac{L}{D} \int_0^D \left| \mathbb{P}_Z \left(Z > t\right) - \mathbb{P}_{Z'} \left(Z' > t\right) \right| dt \\
&\qquad + \frac{L}{D} \int_0^D \left| \mathbb{P}_Z \left(Z > t\right) - \mathbb{P}_{Z'} \left(Z' > t\right) \right| dt \\
&= 2\frac{L}{D} \int_0^D \left| F_{Z'}(t) - F_Z(t) \right| dt \\
&\leq 2L \max_t \left| F_Z(t) - F_{Z'}(t) \right|
\end{aligned}
$$

$\square$

*Proof of Lemma 4.3.* For the variance of any random variable $Z$ with bounded support $[0, D]$, we have

$$\mathbb{V}(Z) = \mathbb{E}(Z^2) - \mathbb{E}(Z)^2.$$

Note that by the definition of expectation,

$$\mathbb{E}(Z^2) = \int_{t^2=0}^{D^2} 1 - F_{Z^2}(t^2) dt^2$$

Then using $dt^2 = 2t dt$ and the fact that $\mathbb{P}(Z^2 \geq t^2) = \mathbb{P}(Z \geq t)$ since $t$ is nonnegative, with this change of variables we have

$$\mathbb{E}(Z^2) = 2 \int_{t=0}^D t \left(1 - F_Z(t)\right) dt.$$

This gives us the following expression for variance:

$$\mathbb{V}(Z) = 2 \int_0^D t(1 - F_Z(t)) dt - \left( \int_0^D (1 - F_Z(t)) dt \right)^2$$

Next, consider a pair of random variables $Z$ and $Z'$ with $F_Z$ and $F_{Z'}$ as their CDF respectively. Therefore,

$$|\mathbb{V}(Z) - \mathbb{V}(Z')| \leq \left|2\int_0^D t(F_Z(t) - F_{Z'}(t))dt\right| + \left|\left(\int_0^D (1 - F_Z(t))dt\right)^2 - \left(\int_0^D (1 - F_{Z'}(t))dt\right)^2\right|$$

$$\leq D^2\|F_Z(t) - F_{Z'}\|_\infty + \left|\int_0^D (F_Z(t) - F_{Z'}(t))dt\right|\left|\int_0^D (1 - F_Z(t))dt + \int_0^D (1 - F_{Z'}(t))dt\right|$$

$$\leq D^2\|F_Z(t) - F_{Z'}\|_\infty + 2D\left|\int_0^D (F_Z(t) - F_{Z'}(t))dt\right|$$

$$\leq D^2\|F_Z(t) - F_{Z'}\|_\infty + 2D^2\|F_Z(t) - F_{Z'}\|_\infty$$

$$= 3D^2\|F_Z - F_{Z'}\|_\infty$$

$\square$

*Proof of Lemma 4.4.* The proof of this lemma follows directly from the definition of Lipschitzness:

$$\left|\sum_{k=1}^K \lambda_k \rho_k(Z) - \sum_{k=1}^K \lambda_k \rho_k(Z')\right| \leq \sum_{k=1}^K \lambda_k |\rho_k(Z) - \rho_k(Z')|$$

$$\leq \|F_Z - F_{Z'}\|_\infty \sum_{k=1}^K \lambda_k L_k.$$

$\square$

# B Proofs for Importance Sampling Estimators (Section 5.1)

## B.1 Proof: Bias and Variance of IS CDF Estimate

*Proof of Lemma 5.1.* We take the expectation of the IS estimator (1) with respect to $\mathbb{P}_\beta$. Then for any $t \in \mathbb{R}$,

$$
\begin{aligned}
\mathbb{E}_{\mathbb{P}_\beta}[\widehat{F}_{\text{IS}}(t)] &= \mathbb{E}_{\mathbb{P}_\beta}\left[\frac{1}{n}\sum_{i=1}^{n} w(A_i, X_i)\mathbb{1}_{\{R_i \leq t\}}\right] \\
&= \mathbb{E}_{\mathbb{P}_\beta}\left[\mathbb{E}_{\mathbb{P}_\beta}\left[\frac{\pi(A|X)}{\beta(A|X)}\mathbb{E}_{\mathbb{P}_\beta}\left[\mathbb{1}_{\{R \leq t\}}|X, A\right]\right]\right] \\
&= \mathbb{E}_{\mathbb{P}}\left[w(A, X)\mathbb{1}_{\{R \leq t\}}\right] \\
&= F(t).
\end{aligned}
$$

Recall that $G(t; X, A) = \mathbb{E}[\mathbb{1}_{\{R \leq t\}}|X, A]$. The variance of the IS estimator is derived using:

$$
\begin{aligned}
\mathbb{V}_{\mathbb{P}_\beta}\left[\widehat{F}_{\text{IS}}(t)\right] &= \frac{1}{n}\mathbb{V}_{\mathbb{P}_\beta}\left[w(A, X)\mathbb{1}_{\{R \leq t\}}\right] \\
&= \frac{1}{n}\mathbb{E}_{\mathbb{P}_\beta}\left[w(A, X)^2\mathbb{V}_{\mathbb{P}_\beta}\left[\mathbb{1}_{\{R \leq t\}}|A, X\right]\right] + \frac{1}{n}\mathbb{V}_{\mathbb{P}_\beta}\left[w(A, X)\mathbb{E}_{\mathbb{P}_\beta}\left[\mathbb{1}_{\{R \leq t\}}|A, X\right]\right] \\
&= \frac{1}{n}\mathbb{E}_{\mathbb{P}_\beta}\left[w(A, X)^2\sigma^2(t; X, A)\right] + \frac{1}{n}\mathbb{V}_{\mathbb{P}_\beta}\left[w(A, X)G(t; X, A)\right] \\
&= \frac{1}{n}\mathbb{E}_{\mathbb{P}_\beta}\left[w(A, X)^2\sigma^2(t; X, A)\right] + \frac{1}{n}\mathbb{V}_{\mathbb{P}_\beta}\left[\mathbb{E}_{\mathbb{P}_\beta}\left[w(A, X)G(t; X, A)|X\right]\right] \\
&\quad + \frac{1}{n}\mathbb{E}_{\mathbb{P}_\beta}\left[\mathbb{V}_{\mathbb{P}_\beta}\left[w(A, X)G(t; X, A)|X\right]\right]
\end{aligned}
$$

where the second equality uses the law of total variance conditioned on actions $A$ and contexts $X$, and the third equality uses the definitions of $\sigma^2$ and $G$. The last equality is another application of the law of total variance conditioning on the context $X$.

$\square$

## B.2 Proof: Error Bound of IS CDF Estimate

*Proof Theorem 5.1.* Define the following function class:

$$
\mathbb{F}(n) := \left\{f(r) := \varrho\frac{1}{n}\mathbb{1}_{\{r \leq t\}} : \forall t \in \mathbb{R}; \forall r \in \mathbb{Q}, \varrho \in \{-1, +1\}\right\}
$$

Note that this is a countable set. Using this definition, we have

$$
\sup_{t \in \mathbb{R}}\left|\widehat{F}_{\text{IS}}(t) - F(t)\right| = \sup_{f \in \mathbb{F}(n)}\left|\left(\sum_{i}^{n}\left(w(A_i, X_i)f(R_i) - \mathbb{E}_{\mathbb{P}_\beta}[w(A_i, X_i)f(R_i)]\right)\right)\right|
$$

Using this equality, for $\lambda > 0$, we have:

$$
\begin{aligned}
&\mathbb{E}_{\mathbb{P}_\beta}\left[\exp\left(\lambda\sup_{t \in \mathbb{R}}\left|\widehat{F}_{\text{IS}}(t) - F(t)\right|\right)\right] \\
&= \mathbb{E}_{\mathbb{P}_\beta}\left[\exp\left(\lambda\sup_{f \in \mathbb{F}(n)}\left|\left(\sum_{i}^{n}\left(w(A_i, X_i)f(R_i) - \mathbb{E}_{\mathbb{P}_\beta}[w(A_i, X_i)f(R_i)]\right)\right)\right|\right)\right] \\
&= \mathbb{E}_{\mathbb{P}_\beta}\left[\exp\left(\lambda\sup_{f \in \mathbb{F}(n)}\left|\left(\mathbb{E}_{\mathbb{P}_\beta}\left[\sum_{i}^{n}\left(w(A_i, X_i)f(R_i) - w(X_i', A_i')f(R_i')\right)\Big|\{X_i, A_i, R_i\}_i^n\right]\right)\right|\right)\right] \\
&\leq \mathbb{E}_{\mathbb{P}_\beta}\left[\exp\left(\lambda\sup_{f \in \mathbb{F}(n)}\left|\mathbb{E}_{\mathbb{P}_\beta}\left[\left(\sum_{i}^{n}\left(w(A_i, X_i)f(R_i) - w(X_i', A_i')f(R_i')\right)\Big|\{X_i, A_i, R_i\}_i^n\right)\right]\right|\right)\right] \\
&\leq \mathbb{E}_{\mathbb{P}_\beta}\left[\exp\left(\lambda\mathbb{E}_{\mathbb{P}_\beta}\left[\sup_{f \in \mathbb{F}(n)}\left|\left(\sum_{i}^{n}\left(w(A_i, X_i)f(R_i) - w(X_i', A_i')f(R_i')\right)\Big|\{X_i, A_i, R_i\}_i^n\right)\right|\right]\right)\right]
\end{aligned}
$$

$$\leq \mathbb{E}_{\mathbb{P}_\beta}\left[\exp\left(\lambda \sup_{f\in\mathbb{F}(n)}\left|\left(\sum_i^n (w(A_i,X_i)f(R_i) - w(X_i',A_i')f(R_i'))\right)\right|\right)\right]$$

$$= \mathbb{E}_{\mathbb{P}_\beta,\mathfrak{R}}\left[\exp\left(\lambda \sup_{f\in\mathbb{F}(n)}\left|\left(\sum_i^n \xi_i(w(A_i,X_i)f(R_i) - w(X_i',A_i')f(R_i'))\right)\right|\right)\right]$$

$$\leq \mathbb{E}_{\mathbb{P}_\beta,\mathfrak{R}}\left[\exp\left(2\lambda \sup_{f\in\mathbb{F}(n)}\left|\left(\sum_i^n \xi_i w(A_i,X_i)f(R_i)\right)\right|\right)\right]$$

$$= \mathbb{E}_{\mathbb{P}_\beta,\mathfrak{R}}\left[\sup_{f\in\mathbb{F}(n)} \exp\left(2\lambda\left|\left(\sum_i^n \xi_i w(A_i,X_i)f(R_i)\right)\right|\right)\right]$$

with $\mathfrak{R}$ a Rademacher measure on a set of Rademacher random variable $\{\xi_i\}$ a Rademacher random variable.

Next, permute the indices $i$ such that $R_1 \leq \ldots R_i \ldots \leq R_n$. Consider a function $f(r) = \frac{1}{n}\varrho\mathbb{1}_{\{r\leq t\}}$. For such a function, $\sum_i^n \xi_i w(A_i,X_i)f(R_i)$ is equal to

- $0$ if $t < \min_i\{R_i\}_i^n$,
- $\frac{1}{n}\varrho \sum_i^j w(A_i,X_i)\xi_i$ when $R_j \leq t < R_{j+1}$ for a $j \in \{1,\ldots,n-1\}$,
- $\frac{1}{n}\varrho \sum_i^n w(A_i,X_i)\xi_i$ otherwise.

Then,

$$\sup_{f\in\mathbb{F}(n)} \exp\left(2\lambda\left|\left(\sum_i^n \xi_i w(A_i,X_i)f(R_i)\right)\right|\right)$$

$$= \max_{\varrho,j}\exp\left(\frac{2\lambda}{n}\varrho\sum_i^j w(A_i,X_i)\xi_i\right)$$

$$= \max_j\left(\exp\left(\frac{2\lambda}{n}\sum_i^j w(A_i,X_i)\xi_i\right)\mathbb{1}_{\{\sum_i^j w(A_i,X_i)\xi_i\geq 0\}}\right.$$

$$\left. + \exp\left(-\frac{2\lambda}{n}\sum_i^j w(A_i,X_i)\xi_i\right)\mathbb{1}_{\{\sum_i^j w(A_i,X_i)\xi_i< 0\}}\right)$$

$$= \max_j\left(\exp\left(\frac{2\lambda}{n}\sum_i^j w(A_i,X_i)\xi_i\right)\mathbb{1}_{\{\sum_i^j w(A_i,X_i)\xi_i\geq 0\}}\right)$$

$$+ \max_j\left(\exp\left(-\frac{2\lambda}{n}\sum_i^j w(A_i,X_i)\xi_i\right)\mathbb{1}_{\{\sum_i^j w(A_i,X_i)\xi_i< 0\}}\right)$$

Which gives us the inequality

$$\mathbb{E}_{\mathbb{P}_\beta}\left[\exp\left(\lambda\sup_{t\in\mathbb{R}}\left|\widehat{F}_{\mathrm{IS}}(t) - F(t)\right|\right)\right] \leq 2\mathbb{E}_{\mathbb{P}_\beta,\mathfrak{R}}\left[\max_j\exp\left(\frac{2\lambda}{n}\sum_i^j w(A_i,X_i)\xi_i\right)\mathbb{1}_{\{\sum_i^j w(A_i,X_i)\xi_i\geq 0\}}\right] \tag{9}$$

Now we are left to bound the right hand side of (9). Using Lemma B.1, for the right hand side of the (9) we have,

$$\mathbb{E}_{\mathbb{P}_\beta,\mathfrak{R}}\left[\exp\left(\frac{2\lambda}{n}\max_j\sum_i^j w(A_i,X_i)\xi_i\right)\mathbb{1}_{\{\max_j\sum_i^j w(A_i,X_i)\xi_i\geq 0\}}\right]$$

$$= \mathbb{P}_\beta \{\max_j \frac{2\lambda}{n} \sum_i^j w(A_i, X_i)\xi_i \geq 0\}$$

$$+ \lambda \int_0^\infty \exp(\lambda t)\mathbb{P}\{\max_j \frac{2\lambda}{n} \sum_i^j w(A_i, X_i)\xi_i \geq t\}dt$$

$$\leq \mathbb{P}_\beta \{\max_j \frac{2\lambda}{n} \sum_i^j w(A_i, X_i)\xi_i \geq 0\}$$

$$+ 2\lambda \int_0^\infty \exp(\lambda t)\mathbb{P}\{\frac{2\lambda}{n} \sum_i w(A_i, X_i)\xi_i \geq t\}dt \tag{10}$$

Note that similarly we have,

$$\mathbb{E}_{\mathbb{P}_\beta, \mathfrak{R}}\left[\exp\left(\frac{2\lambda}{n}\sum_i w(A_i, X_i)\xi_i\right)\mathbb{1}_{\{\sum_i w(A_i, X_i)\xi_i \geq 0\}}\right]$$

$$= \mathbb{P}_\beta\{\frac{2\lambda}{n}\sum_i w(A_i, X_i)\xi_i \geq 0\} + \lambda \int_0^\infty \exp(\lambda t)\mathbb{P}\{\frac{2\lambda}{n}\sum_i w(A_i, X_i)\xi_i \geq t\}dt \tag{11}$$

Putting these two statements, i.e., (10), and (11) together, and applying the result of Lemma B.2, we have,

$$\mathbb{E}_{\mathbb{P}_\beta, \mathfrak{R}}\left[\exp\left(\max_j \frac{2\lambda}{n}\sum_i^j w(A_i, X_i)\xi_i\right)\mathbb{1}_{\{\sum_i^j w(A_i, X_i)\xi_i \geq 0\}}\right]$$

$$\leq \mathbb{P}_\beta\{\max_j \frac{2\lambda}{n}\sum_i^j w(A_i, X_i)\xi_i \geq 0\}$$

$$+ 2\mathbb{E}_{\mathbb{P}_\beta, \mathfrak{R}}\left[\exp\left(\frac{2\lambda}{n}\sum_i w(A_i, X_i)\xi_i\right)\mathbb{1}_{\{\sum_i w(A_i, X_i)\xi_i \geq 0\}}\right]$$

$$- 2\mathbb{P}_\beta\{\frac{2\lambda}{n}\sum_i w(A_i, X_i)\xi_i \geq 0\}$$

$$\leq 2\mathbb{E}_{\mathbb{P}_\beta, \mathfrak{R}}\left[\exp\left(\frac{2\lambda}{n}\sum_i w(A_i, X_i)\xi_i\right)\mathbb{1}_{\{\sum_i w(A_i, X_i)\xi_i \geq 0\}}\right]$$

$$\leq 2\mathbb{E}_{\mathbb{P}_\beta, \mathfrak{R}}\left[\exp\left(\frac{2\lambda}{n}\sum_i w(A_i, X_i)\xi_i\right)\right]$$

Note that $\frac{2}{n}w(A_i, X_i)\xi_i$ is a mean zero random variable with values in $[-\frac{2}{n}w_{\max}, \frac{2}{n}w_{\max}]$. Therefore, it is a sub-Gaussian random variable with sub-Gaussian constant as $\left(\frac{2}{n}\right)^2 w_{\max}^2$. Using this, we have, $\frac{2}{n}\sum_i w(A_i, X_i)\xi_i$ is $\frac{4}{n}w_{\max}^2$ sub-Gaussian random variable. Therefore, we have,

$$\mathbb{E}_{\mathbb{P}_\beta, \mathfrak{R}}\left[\exp\left(\max_j \frac{2\lambda}{n}\sum_i^j w(A_i, X_i)\xi_i\right)\mathbb{1}_{\{\sum_i^j w(A_i, X_i)\xi_i \geq 0\}}\right] \leq 2\mathbb{E}_{\mathbb{P}_\beta, \mathfrak{R}}\left[\exp\left(\frac{2\lambda}{n}\sum_i w(A_i, X_i)\xi_i\right)\right]$$

$$\leq 2\exp\left(\lambda^2 \frac{2}{n}w_{\max}^2\right)$$

Putting this with the (9), we have

$$\mathbb{E}_{\mathbb{P}_\beta}\left[\exp\left(\lambda \sup_{t\in\mathbb{R}}\left|\widehat{F}_{\text{IS}}(t) - F(t)\right|\right)\right] \le 2\mathbb{E}_{\mathbb{P}_\beta,\mathfrak{R}}\left[\exp\left(\max_j \frac{2\lambda}{n}\sum_i^j w(A_i, X_i)\xi_i\right)\mathbb{1}_{\{\sum_i^j w(A_i,X_i)\xi_i\ge 0\}}\right]$$

$$\le 4\exp\left(\lambda^2 \frac{2}{n} w_{\max}^2\right)$$

Using Markov inequality we have

$$\mathbb{P}_\beta\left(\sup_{t\in\mathbb{R}}\left|\widehat{F}_{\text{IS}}(t) - F(t)\right| \ge \epsilon\right) = \mathbb{P}_\beta\left(\exp\left(\lambda \sup_{t\in\mathbb{R}}\left|\widehat{F}_{\text{IS}}(t) - F(t)\right|\right) \ge \exp(\lambda\epsilon)\right)$$

$$\le 4\exp\left(\lambda^2 \frac{2}{n} w_{\max}^2\right)\exp(-\lambda\epsilon)$$

$$= 4\exp\left(\lambda^2 \frac{2}{n} w_{\max}^2 - \lambda\epsilon\right)$$

This holds for any choice of $\lambda > 0$, resulting in

$$\mathbb{P}_\beta\left(\sup_{t\in\mathbb{R}}\left|\widehat{F}_{\text{IS}}(t) - F(t)\right| \ge \epsilon\right) \le \inf_{\lambda>0} 4\exp\left(\lambda^2 \frac{2}{n} w_{\max}^2 - \lambda\epsilon\right) = 4\exp\left(\frac{-n\epsilon^2}{8 w_{\max}^2}\right)$$

Using this, we have

$$\mathbb{P}_\beta\left(\sup_{t\in\mathbb{R}}\left|\widehat{F}_{\text{IS}}(t) - F(t)\right| \le \sqrt{\frac{8 w_{\max}^2}{n}\log\left(\frac{4}{\delta}\right)}\right) \ge 1 - \delta.$$

**Bernstein style:** To bound this $\mathbb{E}_{\mathbb{P}_\beta,\mathfrak{R}}\left[\exp\left(\frac{2\lambda}{n}\sum_i w(A_i, X_i)\xi_i\right)\right]$ now we use Bernstein's. As discussed, the random variable $w(A_i, X_i)\xi_i$ is in $[-w_{\max}, w_{\max}]$. However, if we look at its variance, we have $\mathbb{E}_{\mathbb{P}_\beta,\mathfrak{R}}\left[w(A_i, X_i)^2\xi_i^2\right] = \mathbb{E}_{\mathbb{P}_\beta,\mathfrak{R}}\left[w(A_i, X_i)^2\right]$ which is the second order Rényi divergence $w_2$. Therefore, for $0 < \lambda < \frac{n}{2w_{\max}}$, we have

$$\mathbb{E}_{\mathbb{P}_\beta,\mathfrak{R}}\left[\exp\left(\frac{2\lambda}{n}\sum_i w(A_i, X_i)\xi_i\right)\right] = \prod_i \mathbb{E}_{\mathbb{P}_\beta,\mathfrak{R}}\left[\exp\left(\frac{2\lambda}{n} w(A_i, X_i)\xi_i\right)\right]$$

$$\le \prod_i \exp\left(\frac{\lambda^2 \frac{4w_2}{n^2}}{2\left(1 - \lambda\frac{2}{n} w_{\max}\right)}\right)$$

$$= \exp\left(\frac{n\lambda^2 \frac{4w_2}{n^2}}{2\left(1 - \lambda\frac{2}{n} w_{\max}\right)}\right)$$

Using the Markov inequality, we have,

$$\mathbb{P}_\beta\left(\sup_{t\in\mathbb{R}}\left|\widehat{F}_{\text{IS}}(t) - F(t)\right| \ge \epsilon\right) = 4\exp\left(\frac{n\lambda^2 \frac{4w_2}{n^2}}{2\left(1 - \lambda\frac{2}{n} w_{\max}\right)} - \lambda\epsilon\right)$$

Setting $\lambda = \frac{\epsilon}{\frac{2w_{\max}\epsilon}{n} + n\frac{4w_2}{n^2}}$, we have,

$$\mathbb{P}_\beta\left(\sup_{t\in\mathbb{R}}\left|\widehat{F}_{\text{IS}}(t) - F(t)\right| \ge \epsilon\right) \le 4\exp\left(\frac{-\epsilon^2}{2\left(\frac{2}{n} w_{\max}\epsilon + n\frac{4w_2}{n^2}\right)}\right)$$

$$= 4 \exp\left(\frac{-n\epsilon^2}{4w_{\max}\epsilon + 8w_2}\right)$$

which results in,

$$\mathbb{P}_\beta\left(\sup_{t\in\mathbb{R}}\left|\widehat{F}_{\text{IS}}(t) - F(t)\right| \leq \frac{4w_{\max}\log(\frac{4}{\delta})}{n} + 2\sqrt{\frac{2w_2\log(\frac{4}{\delta})}{n}}\right) \geq 1 - \delta.$$

Finally, we note that since $\sup_t |\widehat{F}_{\text{IS-clip}}(t) - F(t)| \leq \sup_t |\widehat{F}_{\text{IS}}(t) - F(t)|$, the above results for $\widehat{F}_{\text{IS}}$ also hold for $\widehat{F}_{\text{IS-clip}}$. $\qquad\square$

**Auxiliary Lemmas**

**Lemma B.1.** *For any random variable $X$, with probability measure $\mathbb{P}$, we have*

$$\mathbb{E}\left[\exp(\lambda X)\mathbb{1}_{\{X\geq 0\}}\right] = \mathbb{P}\{X \geq 0\} + \lambda \int_0^\infty \exp(\lambda t)\mathbb{P}\{X \geq t\}dt.$$

*Proof.* for any random variable $X$, with probability measure $\mathbb{P}$, we have

$$\mathbb{E}\left[\exp(\lambda X)\mathbb{1}_{\{X\geq 0\}}\right] = \mathbb{E}\left[\left(\exp(0) + \int_0^X \lambda\exp(\lambda t)dt\right)\mathbb{1}_{\{X\geq 0\}}\right]$$

$$= \mathbb{E}\left[\mathbb{1}_{\{X\geq 0\}}\exp(0)\right] + \mathbb{E}\left[\mathbb{1}_{\{X\geq 0\}}\lambda\int_0^X \exp(\lambda t)\mathbb{1}_{\{X\geq 0\}}dt\right]$$

$$= \mathbb{P}\{X \geq 0\} + \mathbb{E}\left[\lambda\int_0^X \exp(\lambda t)\mathbb{1}_{\{X\geq 0\}}dt\right]$$

$$= \mathbb{P}\{X \geq 0\} + \lambda\int_0^\infty \exp(\lambda t)\mathbb{P}\{X \geq t\}dt. \tag{12}$$

$\qquad\square$

**Lemma B.2.** *For $\gamma > 0$, we have,*

$$\mathbb{P}_\beta\left[\max_j \sum_i^j w(A_i, X_i)\xi_i \geq \gamma\right] \leq 2\mathbb{P}_\beta\left[\sum_i^n w(A_i, X_i)\xi_i \geq \gamma\right] \tag{13}$$

*Proof.* Consider events $E_j := \{\sum_i^j w(A_i, X_i)\xi_i \geq \gamma, \sum_i^l w(A_i, X_i)\xi_i < \gamma, \forall l < j\}$ with $E_0 := \emptyset$. Using these definitions, we have,

$$\{\max_j \sum_i^j w(A_i, X_i)\xi_i \geq \gamma\} \subset \bigcup_j E_j$$

Also,

$$\bigcup_j\left(E_j\bigcap\{\sum_{i>j} w(A_i, X_i)\xi_i \geq 0\}\right) \subset \{\sum_i w(A_i, X_i)\xi_i \geq \gamma\}$$

Also note that

$$\mathbb{P}_\beta\left[\sum_{i>j} w(A_i, X_i)\xi_i \geq 0\right] \geq \frac{1}{2}$$

since this quantity is mean zero and symmetric. Also note that the event $\sum_{i>j} w(A_i, X_i)\xi_i$ is independent of $E_j$.

Using these, we have,

$$\mathbb{P}_\beta \left[ E_j \bigcap \{\sum_{i>j} w(A_i, X_i)\xi_i \geq 0\} \right] = \mathbb{P}_\beta \left[ E_j \right] \mathbb{P}_\beta \left[ \{\sum_{i>j} w(A_i, X_i)\xi_i \geq 0\} \right] \geq \frac{\mathbb{P}_\beta \left[ E_j \right]}{2}$$

As a result we have,

$$\mathbb{P}_\beta \left[ \sum_i w(A_i, X_i)\xi_i \geq \gamma \right] \geq \mathbb{P}_\beta \left[ \bigcup_j \left( E_j \bigcap \{\sum_{i>j} w(A_i, X_i)\xi_i \geq 0\} \right) \right]$$

$$= \sum_j \mathbb{P}_\beta \left[ E_j \bigcap \{\sum_{i>j} w(A_i, X_i)\xi_i \geq 0\} \right]$$

$$\geq \sum_j \frac{\mathbb{P}_\beta \left[ E_j \right]}{2}$$

$$\geq \frac{\mathbb{P}_\beta \left[ \{\max_j \sum_i^j w(A_i, X_i)\xi_i \geq \gamma\} \right]}{2}$$

which concludes the statement. □

# C  Proofs for Doubly Robust Estimators (Section 5.3)

## C.1  Proof: Bias and Variance of DR CDF Estimate

*Proof of Lemma 5.2.* The expectation of the DR estimator (22) is as follows:

$$
\begin{aligned}
\mathbb{E}_{\mathbb{P}_\beta}\left[\widehat{F}_{\mathrm{DR}}(t)\right] &= \mathbb{E}_{\mathbb{P}_\beta}\left[w(A,X)\mathbb{1}_{\{R\leq t\}}\right] + \mathbb{E}_{\mathbb{P}_\beta}\left[\overline{G}(t;X,\pi) - w(A,X)\overline{G}(t;X,A)\right] \\
&= F(t) + \mathbb{E}_{\mathbb{P}_\beta}\left[\overline{G}(t;X,\pi) - \mathbb{E}_{\mathbb{P}_\beta}[w(A,X)\overline{G}(t;X,A)|X]\right] \\
&= F(t) + \mathbb{E}_{\mathbb{P}_\beta}\left[\overline{G}(t;X,\pi) - \overline{G}(t;X,\pi)\right] \\
&= F(t).
\end{aligned}
$$

Next, we derive the variance.

$$
\begin{aligned}
\mathbb{V}_{\mathbb{P}_\beta}\left[\widehat{F}_{\mathrm{DR}}(t)\right] &= \frac{1}{n}\mathbb{V}_{\mathbb{P}_\beta}\left[w(A,X)\left(\mathbb{1}_{\{R\leq t\}} - \overline{G}(t;X,A)\right) + \overline{G}(t;X,\pi)\right] \\
&= \frac{1}{n}\mathbb{E}_{\mathbb{P}_\beta}\left[w(A,X)^2\sigma^2(t;X,A)\right] \\
&\quad + \frac{1}{n}\mathbb{V}_{\mathbb{P}_\beta}\left[w(A,X)\left(G(t;X,A) - \overline{G}(t;X,A)\right) + \overline{G}(t;X,\pi)\right] \\
&= \frac{1}{n}\mathbb{E}_{\mathbb{P}_\beta}\left[w(A,X)^2\sigma^2(t;X,A)\right] + \frac{1}{n}\mathbb{V}_{\mathbb{P}_\beta}\left[\mathbb{E}_{\mathbb{P}_\beta}\left[w(A,X)G(t;X,A)|X\right]\right] \\
&\quad + \frac{1}{n}\mathbb{E}_{\mathbb{P}_\beta}\left[\mathbb{V}_{\mathbb{P}_\beta}\left[w(A,X)\left(G(t;X,A) - \overline{G}(t;X,A)\right)|X\right]\right]
\end{aligned}
$$

The first equality follows from applying the law of total variance, noting that the variance $\mathbb{V}_{\mathbb{P}_\beta}\left[\overline{G}(t;X,A)|X,A\right] = 0$, and using the definitions of $G$ and $\sigma^2$. The second equality again applies the law of total variance. $\qquad\square$

## C.2  Proof: Error Bound of DR CDF Estimate

*Proof of Theorem 5.2.* Recall that the DR estimator $\widehat{F}_{\mathrm{DR}}(t)$ is defined as

$$
\widehat{F}_{\mathrm{DR}}(t) = \frac{1}{n}\sum_{i=1}^n w(a_i,x_i)\left(\mathbb{1}_{\{r_i\leq t\}} - \overline{G}(t;x_i,a_i)\right) + \overline{G}(t;x_i,\pi)
$$

where $\overline{G}(t;x,\pi) = \mathbb{E}_{A\sim\pi}\left[\overline{G}(t;x,A)\right]$. We can decompose the error of the DR estimator as:

$$
\begin{aligned}
&\mathbb{E}_{\mathbb{P}_\beta}\left[\sup_t|\widehat{F}_{\mathrm{DR}}(t) - F(t)|\right] \\
&= \mathbb{E}_{\mathbb{P}_\beta}\left[\sup_t\left|\left(\frac{1}{n}\sum_{i=1}^n w(A_i,X_i)\left(\mathbb{1}_{\{R_i\leq t\}} - \overline{G}(t;X_i,A_i)\right) + \overline{G}(t;X_i,\pi)\right) - F(t)\right|\right] \\
&\leq \mathbb{E}_{\mathbb{P}_\beta}\left[\sup_t\left(\left|\frac{1}{n}\sum_{i=1}^n w(A_i,X_i)\mathbb{1}_{\{R_i\leq t\}} - F(t)\right| + \left|\frac{1}{n}\sum_{i=1}^n \overline{G}(t;X_i,\pi) - w(A_i,X_i)\overline{G}(t;X_i,A_i)\right|\right)\right] \\
&\leq \mathbb{E}_{\mathbb{P}_\beta}\left[\sup_t\left|\frac{1}{n}\sum_{i=1}^n w(A_i,X_i)\mathbb{1}_{\{R_i\leq t\}} - F(t)\right| + \sup_t\left|\frac{1}{n}\sum_{i=1}^n \overline{G}(t;X_i,\pi) - w(A_i,X_i)\overline{G}(t;X_i,A_i)\right|\right].
\end{aligned}
$$

We have already bounded the first term in Theorem 5.1, and Lemma C.1 bounds the second term. Then in total, we have

$$
\mathbb{P}_\beta\left(\sup_t\left|\widehat{F}_{\mathrm{DR}}(t) - F(t)\right| \geq \sqrt{\frac{8w_{max}^2}{n}\log\left(\frac{4}{\delta}\right)} + \sqrt{\frac{32w_{max}^2}{n}\log\left(\frac{(2n)^{1/2}}{w_{max}\delta}\right)}\right) \leq 2\delta
$$

Simplifying,

$$
\mathbb{P}_\beta\left(\sup_t\left|\widehat{F}_{DR}(t) - F(t)\right| \geq \sqrt{\frac{72w_{max}^2}{n}\log\left(\frac{4n^{1/2}}{\delta}\right)}\right) \leq 2\delta \tag{14}
$$

which gives us our error bound for the DR estimator $\widehat{F}_{\text{DR}}$.

As mentioned previously, however, $\widehat{F}_{\text{DR}}$ may not be monotone, and in practice we must use a monotone transformation of the estimator. Consider a monotone transformation $\mathcal{M}$ of $\widehat{F}_{\text{DR}}$ that is a simple accumulation function, e.g. $\forall t$,

$$\mathcal{M}\left(\widehat{F}_{DR}(t)\right) = \max_{t' \leq t} \widehat{F}_{DR}(t')$$

Now we want to bound the error between the monotonized estimate $\mathcal{M}\left(\widehat{F}_{DR}(t)\right)$ and $F$. Using our error bound in (14), let $\epsilon = \sqrt{\frac{72 w_{max}^2}{n} \log\left(\frac{8 n^{1/2}}{\delta}\right)}$. Then with probability at least $1 - \delta$, for all $t \in \mathbb{R}$,

$$\max_t |\widehat{F}_{\text{DR}}(t) - F(t)| \leq \epsilon.$$

On this event, $\forall t$ there exists some $t' \leq t$ for which

$$\max_{t' \leq t} \widehat{F}_{DR}(t') - F(t) = \widehat{F}_{DR}(t') - F(t)$$

Using the fact that $F$ is monotone thus $F(t') \leq F(t)$, when $\widehat{F}_{DR}(t') \geq F(t)$ we have

$$\widehat{F}_{DR}(t') - F(t) \leq \widehat{F}_{DR}(t') - F(t') \leq \epsilon$$

Similarly, when $\widehat{F}_{DR}(t') \leq F(t)$,

$$F(t) - \widehat{F}_{DR}(t') \leq F(t) - \widehat{F}_{DR}(t) \leq \epsilon$$

Putting these two inequalities together, we have

$$\max_t \left|\mathcal{M}\left(\widehat{F}_{DR}\right)(t) - F(t)\right| \leq \epsilon.$$

The theorem statement, which applies to the clipped monotone transformation, follows from the fact that

$$\max_t \left|\min\left\{\mathcal{M}\left(\widehat{F}_{DR}\right)(t), 1\right\} - F(t)\right| \leq \max_t \left|\mathcal{M}\left(\widehat{F}_{DR}\right)(t) - F(t)\right|.$$

$\square$

**Lemma C.1.** *Let $\overline{G}(t; x, a)$ be a valid conditional CDF for all $x \in \mathcal{X}, a \in \mathcal{A}$, and let $w : \mathcal{A} \times \mathcal{X} \to \mathbb{R}$ be the importance sampling weights. Then for $\delta \in (0, 1]$,*

$$\mathbb{P}_\beta \left( \sup_t \left| \frac{1}{n} \sum_{i=1}^n \overline{G}(t; X_i, \pi) - \frac{1}{n} \sum_{i=1}^n w(A_i, X_i) \overline{G}(t; X_i, A_i) \right| \geq \sqrt{\frac{32 w_{max}^2}{n} \log \frac{(2n)^{1/2}}{w_{max} \delta}} \right) \leq \delta.$$

*where $\overline{G}(t; x, \pi) = \mathbb{E}_{\mathbb{P}}[\overline{G}(t; x, A) | x]$.*

*Proof.* Since $\overline{G}$ is a valid CDF, we apply Lemma C.2 to $\overline{G}$. Consider a function of the form

$$\overline{\zeta}(t; s^1, ..., s^m) = \frac{1}{m} \sum_{j=1}^m \mathbb{1}_{\{s^i \leq t\}}$$

The function $\overline{\zeta}$ can be seen as a stepwise CDF function, where each step is $1/m$ and occurs at points $\{s^j\}_{j=1}^m$.

Lemma C.2 approximates $\overline{G}$ using such $1/m$-stepwise CDFs. For each context $x$ and action $a$, let $s_{x,a}^1, ..., s_{x,a}^m \in \mathbb{Q}^m$ be the points chosen according to the deterministic procedure in Lemma C.2, such that the following inequality holds:

$$\sup_t \left|\overline{G}(t; x, a) - \overline{\zeta}\left(t; \{s_{x,a}^j\}_{j=1}^m\right)\right| \leq \frac{1}{2m}. \tag{15}$$

Next, consider the class of functions

$$\mathcal{G}(m) := \left\{ \zeta(s^1, ..., s^m) := \frac{1}{m} \varrho \sum_{j=1}^{m} \mathbb{1}_{\{s^j \leq t\}} : \forall t \in \mathbb{R}, \varrho \in \{-1, +1\}; \{s^j\}_{j=1}^{m} \in \mathbb{Q}^m \right\}$$

$$\mathcal{G}(m) := \left\{ \zeta(\cdot; s^1, ..., s^m) : \mathbb{R} \to [0, 1] := \frac{1}{m} \varrho \sum_{j=1}^{m} \mathbb{1}_{\{s^j \leq t\}} : \varrho \in \{-1, +1\}; \{s^j\}_{j=1}^{m} \in \mathbb{Q}^m \right\}$$

Note that, $\overline{\zeta}$ is a subset of the function class $\mathcal{G}(m)$, e.g. $\overline{\zeta}\left(t; \{s_{x,a}^j\}_{j=1}^{m}\right) \in \mathcal{G}(m)$.

Then our problem becomes

$$\sup_t \left| \frac{1}{n} \sum_{i=1}^{n} w(A_i, X_i) \overline{G}(t; X_i, A_i) - \frac{1}{n} \sum_{i=1}^{n} \overline{G}(t; X_i, \pi) \right|$$

$$= \sup_t \left| \frac{1}{n} \sum_{i=1}^{n} w(A_i, X_i) \overline{G}(t; X_i, A_i) - \frac{1}{n} \sum_{i=1}^{n} \mathbb{E}_{\mathbb{P}} \left[ \overline{G}(t; X_i, A) | X_i \right] \right|$$

$$= \sup_t \left| \frac{1}{n} \sum_{i=1}^{n} w(A_i, X_i) \overline{G}(t; X_i, A_i) - \frac{1}{n} \sum_{i=1}^{n} \mathbb{E}_{\mathbb{P}_\beta} \left[ w(X_i, A) \overline{G}(t; X_i, A) | X_i \right] \right|$$

$$\leq \sup_t \left| \frac{1}{n} \sum_{i=1}^{n} w(A_i, X_i) \overline{\zeta}\left(t; \{s_{X_i, A_i}^j\}_{j=1}^{m}\right) - \frac{1}{n} \sum_{i=1}^{n} \mathbb{E}_{\mathbb{P}_\beta} \left[ w(A, X_i) \overline{\zeta}\left(t; \{s_{X_i, A}^j\}_{j=1}^{m}\right) \Big| X_i \right] \right| + \frac{1}{m}$$

$$\leq \sup_{\zeta \in \mathcal{G}(m)} \left| \frac{1}{n} \sum_{i=1}^{n} w(A_i, X_i) \zeta(\{s_{X_i, A_i}^j\}_{j=1}^{m}) - \frac{1}{n} \sum_{i=1}^{n} \mathbb{E}_{\mathbb{P}_\beta} \left[ w(A, X_i) \zeta(\{s_{X_i, A}^j\}_{j=1}^{m}) \Big| X_i \right] \right| + \frac{1}{m}$$

where the second line uses the definition of $\overline{G}(t; X_i, \pi)$, the third line uses a change of measure through the importance sampling weight $w$, the fourth line uses (C.2), and the last line uses the fact that, conditioned on $\{s_{x,a}^j\}_{j=1}^{m}$, the function $\overline{\zeta}$ is a member of $\mathcal{G}(m)$.

We can now upper bound the RHS. Going forward, we refer to $\zeta(\{s_{X,A}^j\}_{j=1}^{m})$ as $\zeta(X, A)$ for short. Then for $\lambda > 0$ we have:

$$\mathbb{E}_{\mathbb{P}_\beta} \left[ \exp\left( \lambda \sup_{\zeta \in \mathcal{G}(m)} \left( \frac{1}{n} \sum_{i=1}^{n} w(A_i, X_i) \zeta(X_i, A_i) - \frac{1}{n} \sum_{i=1}^{n} \mathbb{E}_{\mathbb{P}_\beta} \left[ w(A, X_i) \zeta(X_i, A) \Big| X_i \right] \right) \right) \right]$$

$$= \mathbb{E}_{\mathbb{P}_\beta} \left[ \exp\left( \lambda \sup_{\zeta \in \mathcal{G}(m)} \left( \frac{1}{n} \sum_{i=1}^{n} \mathbb{E}_{\mathbb{P}_\beta} \left[ w(A_i, X_i) \zeta(X_i, A_i) - w(A_i', X_i) \zeta(X_i, A_i') \Big| \{X_i, A_i\}_{i=1}^{n} \right] \right) \right) \right]$$

$$\leq \mathbb{E}_{\mathbb{P}_\beta} \left[ \exp\left( \lambda \mathbb{E}_{\mathbb{P}_\beta} \left[ \sup_{\zeta \in \mathcal{G}(m)} \frac{1}{n} \sum_{i=1}^{n} (w(A_i, X_i) \zeta(X_i, A_i) - w(A_i', X_i) \zeta(X_i, A_i')) \Big| \{X_i, A_i\}_{i=1}^{n} \right] \right) \right]$$

$$\leq \mathbb{E}_{\mathbb{P}_\beta} \left[ \exp\left( \lambda \sup_{\zeta \in \mathcal{G}(m)} \frac{1}{n} \sum_{i=1}^{n} (w(A_i, X_i) \zeta(X_i, A_i) - w(A_i', X_i) \zeta(X_i, A_i')) \right) \right]$$

$$\leq \mathbb{E}_{\mathbb{P}_\beta, \Re} \left[ \exp\left( 2\lambda \sup_{\zeta \in \mathcal{G}(m)} \frac{1}{n} \sum_{i=1}^{n} \xi_i w(A_i, X_i) \zeta(X_i, A_i) \right) \right]$$

$$= \mathbb{E}_{\mathbb{P}_\beta, \Re} \left[ \sup_{\zeta \in \mathcal{G}(m)} \exp\left( 2\lambda \frac{1}{n} \sum_{i=1}^{n} \xi_i w(A_i, X_i) \zeta(X_i, A_i) \right) \right]$$

$$= \mathbb{E}_{\mathbb{P}_\beta, \Re} \left[ \sup_{t, \varrho} \exp\left( 2\lambda \frac{\varrho}{nm} \sum_{j=1}^{m} \sum_{i=1}^{n} \xi_i w(A_i, X_i) \mathbb{1}_{\{s_{X_i, A_i}^j \leq t\}} \right) \right] \tag{16}$$

where $\{A'\}_i^n$ are the ghost variables, the second to last inequality uses symmetrization (Lemma C.3), and the last line uses the definition of $\zeta(X_i, A_i) = \zeta(s_{X_i, A_i}^1, ..., s_{X_i, A_i}^m)$.

Now, for each $j$, permute the indices $i$ such that $s^j_{X_{j(1)}, A_{j(1)}} \leq \ldots \leq s^j_{X_{j(i)}, A_{j(i)}} \leq \ldots \leq s^j_{X_{j(n)}, A_{j(n)}}$. Then, for a given $j$, consider the function

$$\sum_{i=1}^{n} \xi_{j(i)} w(A_{j(i)}, X_{j(i)}) \mathbb{1}_{\{s^j_{X_{j(i)}, A_{j(i)}} \leq t\}},$$

which equals

1. $0$ if $t < s^j_{X_{j(1)}, A_{j(1)}}$,

2. $\varrho \sum_{i=1}^{k} w(A_{j(i)}, X_{j(i)}) \xi_{j(i)}$ if there exists $k \in \{1, \ldots, n-1\}$ such that $s^j_{X_{j(k)}, A_{j(k)}} \leq t \leq s^j_{X_{j(k)+1}, A_{j(k)+1}}$,

3. $\varrho \sum_{i=1}^{n} w(A_{j(i)}, X_{j(i)}) \xi_{j(i)}$ otherwise.

Then the RHS of (19) equals

$$\mathbb{E}_{\mathbb{P}_\beta, \mathfrak{R}} \left[ \sup_{t, \varrho} \exp \left( 2\lambda \frac{\varrho}{nm} \sum_{j=1}^{m} \sum_{i=1}^{n} \xi_i w(A_i, X_i) \mathbb{1}_{\{s^j_{X_i, A_i} \leq t\}} \right) \right]$$

$$= \mathbb{E}_{\mathbb{P}_\beta, \mathfrak{R}} \left[ \max_{k, \varrho} \exp \left( 2\lambda \frac{\varrho}{nm} \sum_{j=1}^{m} \sum_{i=1}^{k} \xi_{j(i)} w(A_{j(i)}, X_{j(i)}) \right) \right]$$

$$\leq \mathbb{E}_{\mathbb{P}_\beta, \mathfrak{R}} \left[ \max_{j, k, \varrho} \exp \left( 2\lambda \frac{\varrho}{n} \sum_{i=1}^{k} \xi_{j(i)} w(A_{j(i)}, X_{j(i)}) \right) \right].$$

Further, we have that

$$\max_{j, k, \varrho} \exp \left( 2\lambda \frac{\varrho}{n} \sum_{i=1}^{k} \xi_{j(i)} w(A_{j(i)}, X_{j(i)}) \right)$$

$$= \max_{j, k} \left( \exp \left( \frac{2\lambda}{n} \sum_{i}^{k} w(A_{j(i)}, X_{j(i)}) \xi_{j(i)} \right) \mathbb{1}_{\{\sum_i^k w(A_{j(i)}, X_{j(i)}) \xi_{j(i)} \geq 0\}} \right.$$

$$\left. + \exp \left( -\frac{2\lambda}{n} \sum_{i}^{k} w(A_{j(i)}, X_{j(i)}) \xi_{j(i)} \right) \mathbb{1}_{\{\sum_i^k w(a_{j(i)}, x_{j(i)}) \xi_{j(i)} < 0\}} \right)$$

$$\leq 2 \max_{j, k} \exp \left( \frac{2\lambda}{n} \sum_{i}^{k} w(A_{j(i)}, X_{j(i)}) \xi_{j(i)} \right) \mathbb{1}_{\{\sum_i^k w(A_{j(i)}, X_{j(i)}) \xi_{j(i)} \geq 0\}}.$$

Putting it together, we have that

$$\mathbb{E}_{\mathbb{P}_\beta} \left[ \exp \left( \lambda \sup_{\zeta \in \mathcal{G}(m)} \left| \frac{1}{n} \sum_{i=1}^{n} w(A_i, X_i) \zeta(X_i, A_i) - \frac{1}{n} \sum_{i=1}^{n} \mathbb{E}_{A \sim \pi(\cdot | X_i)} \left[ \zeta(X_i, A) | X_i \right] \right| \right) \right]$$

$$\leq 2 \mathbb{E}_{\mathbb{P}_\beta, \mathfrak{R}} \left[ \max_{j, k} \exp \left( \frac{2\lambda}{n} \sum_{i}^{k} w(A_{j(i)}, X_{j(i)}) \xi_{j(i)} \right) \mathbb{1}_{\{\sum_i^k w(A_{j(i)}, X_{j(i)}) \xi_{j(i)} \geq 0\}} \right] \quad (17)$$

Now we are left to bound the RHS of (17). Using Lemma B.1,

$$\mathbb{E}_{\mathbb{P}_\beta, \mathfrak{R}} \left[ \max_{j, k} \exp \left( \frac{2\lambda}{n} \sum_{i}^{k} w(A_{j(i)}, X_{j(i)}) \xi_{j(i)} \right) \mathbb{1}_{\{\max_k \sum_i^k w(A_{j(i)}, X_{j(i)}) \xi_{j(i)} \geq 0\}} \right]$$

$$\leq \mathbb{P}_\beta \left( \max_k \frac{2\lambda}{n} \sum_i^k w(A_{j(i)}, X_{j(i)}) \xi_{j(i)} \geq 0 \right) + 2\lambda \sum_j \int_0^\infty \exp(\lambda t) \mathbb{P} \left( \frac{2\lambda}{n} \sum_{i=1}^{n} w(A_{j(i)}, X_{j(i)}) \xi_{j(i)} \geq t \right) dt.$$

Similarly, for any $j$, we have

$$\mathbb{E}_{\mathbb{P}_\beta, \mathfrak{R}} \left[ \exp \left( \frac{2\lambda}{n} \sum_i^n w(A_{j(i)}, X_{j(i)}) \xi_{j(i)} \right) \mathbb{1}_{\{\sum_i^k w(A_{j(i)}, X_{j(i)}) \xi_{j(i)} \geq 0\}} \right]$$

$$= \mathbb{P}_\beta \left( \frac{2\lambda}{n} \sum_i^n w(A_{j(i)}, X_{j(i)}) \xi_{j(i)} \geq 0 \right) + \lambda \int_0^\infty \exp(\lambda t) \mathbb{P} \left( \frac{2\lambda}{n} \sum_{i=1}^n w(A_{j(i)}, X_{j(i)}) \xi_{j(i)} \geq t \right) dt$$

Putting these two together, we have

$$\mathbb{E}_{\mathbb{P}_\beta, \mathfrak{R}} \left[ \max_{j,k} \exp \left( \frac{2\lambda}{n} \sum_i^k w(A_{j(i)}, X_{j(i)}) \xi_{j(i)} \right) \mathbb{1}_{\{\sum_i^k w(A_{j(i)}, X_{j(i)}) \xi_{j(i)} \geq 0\}} \right]$$

$$\leq \sum_j \mathbb{P}_\beta \left( \max_k \frac{2\lambda}{n} \sum_i^k w(A_{j(i)}, X_{j(i)}) \xi_{j(i)} \geq 0 \right) - 2 \sum_j \mathbb{P}_\beta \left( \frac{2\lambda}{n} \sum_i^n w(A_{j(i)}, X_{j(i)}) \xi_{j(i)} \geq 0 \right)$$

$$+ 2 \sum_j \mathbb{E}_{\mathbb{P}_\beta, \mathfrak{R}} \left[ \exp \left( \frac{2\lambda}{n} \sum_i^n w(A_{j(i)}, X_{j(i)}) \xi_{j(i)} \right) \mathbb{1}_{\{\sum_i^n w(A_{j(i)}, X_{j(i)}) \xi_{j(i)} \geq 0\}} \right]$$

$$\leq 2 \sum_j \mathbb{E}_{\mathbb{P}_\beta, \mathfrak{R}} \left[ \exp \left( \frac{2\lambda}{n} \sum_i^n w(A_{j(i)}, X_{j(i)}) \xi_{j(i)} \right) \mathbb{1}_{\{\sum_i^n w(A_{j(i)}, X_{j(i)}) \xi_{j(i)} \geq 0\}} \right]$$

$$\leq 2m \mathbb{E}_{\mathbb{P}_\beta, \mathfrak{R}} \left[ \exp \left( \frac{2\lambda}{n} \sum_i^n w(A_{j(i)}, X_{j(i)}) \xi_{j(i)} \right) \right]$$

$$\leq 2m \exp \left( \frac{2\lambda^2 w_{max}^2}{n} \right)$$

where the last inequality uses the fact that $\xi$ is a Rademacher random variable, and $w(A, X) \leq w_{max}$. Finally, using Markov's inequality,

$$\mathbb{P}_\beta \left( \sup_t \left| \frac{1}{n} \sum_{i=1}^n \overline{G}(t; X_i, \pi) - w(A_i, X_i) \overline{G}(t; X_i, A_i) \right| \geq \epsilon + \frac{1}{m} \right)$$

$$\leq \mathbb{P}_\beta \left( \exp \left( \lambda \sup_t \left| \frac{1}{n} \sum_{i=1}^n w(A_i, X_i) \zeta(X_i, A_i) - \frac{1}{n} \sum_{i=1}^n \mathbb{E}_{\mathbb{P}_\beta}[\zeta(X_i, A) | X_i] \right| \right) \geq \exp(\lambda \epsilon) \right)$$

$$\leq 4m \exp \left( \frac{2\lambda^2 w_{max}^2}{n} - \lambda \epsilon \right)$$

Because this holds for any $\lambda > 0$, we can minimize the RHS over $\lambda$:

$$\mathbb{P}_\beta \left( \sup_t \left| \frac{1}{n} \sum_{i=1}^n \overline{G}(t; X_i, \pi) - w(A_i, X_i) \overline{G}(t; X_i, A_i) \right| \geq \epsilon + \frac{1}{m} \right) \leq \inf_{\lambda > 0} 4m \exp \left( \frac{2\lambda^2 w_{max}^2}{n} - \lambda \epsilon \right)$$

$$= 4m \exp \left( \frac{-n\epsilon}{8 w_{max}^2} \right).$$

Then we have

$$\mathbb{P}_\beta \left( \sup_t \left| \frac{1}{n} \sum_{i=1}^n \overline{G}(t; X_i, \pi) - w(A_i, X_i) \overline{G}(t; X_i, A_i) \right| \geq \sqrt{\frac{8 w_{max}^2}{n} \log \frac{4m}{\delta}} + \frac{1}{m} \right) \leq \delta.$$

Setting $m = \sqrt{n / 8 w_{max}^2}$ gives the theorem statement:

$$\mathbb{P}_\beta \left( \sup_t \left| \frac{1}{n} \sum_{i=1}^n \overline{G}(t; X_i, \pi) - w(A_i, X_i) \overline{G}(t; X_i, A_i) \right| \geq \sqrt{\frac{32 w_{max}^2}{n} \log \frac{(2n)^{1/2}}{w_{max} \delta}} \right) \leq \delta.$$

$\square$

**Auxiliary Lemmas**

**Lemma C.2.** *For any $\zeta$, a non-decreasing function with support $[0, D]$, there exists $m$ points $s^1....s^m \in \mathbb{Q}^m$ such that for a function of the form,*

$$\overline{\zeta}(t; s^1, ..., s^m) = \frac{1}{m} \sum_{j=1}^{m} \mathbb{1}_{\{s^j \leq t\}}, \quad \forall t \in \mathbb{R}$$

*the following inequality holds:*

$$\|\zeta - \overline{\zeta}\|_\infty \leq \frac{1}{2m}.$$

*Proof of Lemma C.2.* Uniformly partition the interval $[0, D]$ to $m$ partitions, with partition points $\{\frac{j}{D}\}_{j=0}^m$. We construct the set $\{s_j\}_{j=1}^m$ using the following procedure. For any $j \in \{1, \ldots, m\}$ and the corresponding partition point $\frac{j-1}{D}$, let $s^j \in \mathbb{Q}$ be a point such that either $\lim_{t \to s^j_-} \zeta(t) = \frac{j-1}{m} + \frac{1}{2m}$ or $\lim_{t \to s^j_+} \zeta(t) = \frac{j-1}{m} + \frac{1}{2m}$ (e.g., as illustrated in Figure 4). Then for any $t$, $\overline{\zeta}(t)$ is $\frac{1}{2m}$-close to $\zeta(t)$. $\square$

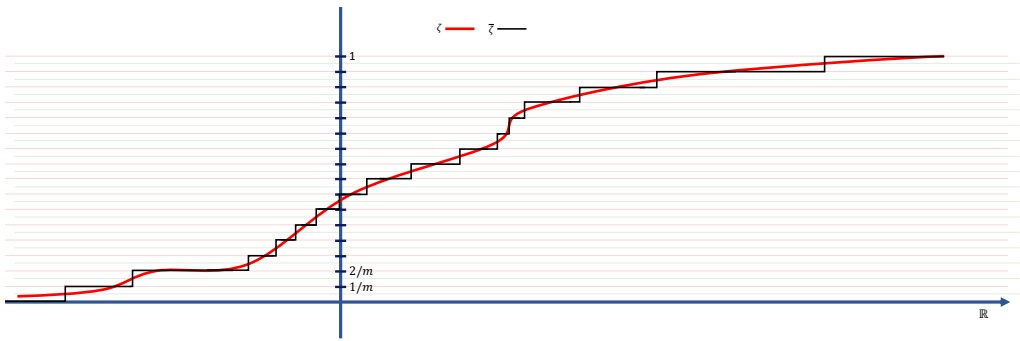

Figure 4: Approximating monotonic function $\zeta$ with $\overline{\zeta}$ when the support is $[0, 1]$ .

**Lemma C.3.** *For the function class $\mathcal{G}$ defined in Appendix C.2, we have for any $\lambda > 0$ that*

$$\mathbb{E}_{\mathbb{P}_\beta} \left[ \exp \left( \lambda \sup_{\zeta \in \mathcal{G}(m)} \frac{1}{n} \sum_{i=1}^{n} (w(A_i, X_i)\zeta(X_i, A_i) - w(A'_i, X_i)\zeta(X_i, A'_i)) \right) \right]$$

$$\leq \mathbb{E}_{\mathbb{P}_\beta, \mathfrak{R}} \left[ \sup_{\zeta \in \mathcal{G}(m)} \exp \left( 2\lambda \frac{1}{n} \sum_{i=1}^{n} \xi_i w(A_i, X_i)\zeta(X_i, A_i) \right) \right]$$

*where contexts and actions $X, A, A' \sim \mathbb{P}_\beta$, and Rademacher random variables $\xi_i \sim \mathfrak{R}$.*

*Proof.* For each $i = 1, ..., n$, and let $\xi_i$ be i.i.d. Rademacher random variables. Set

$$A_i^+ = \begin{cases} A_i, & \text{if } \xi_i = 1 \\ A'_i, & \text{if } \xi_i = -1 \end{cases}$$

$$A_i^- = \begin{cases} A'_i, & \text{if } \xi_i = 1 \\ A_i, & \text{if } \xi_i = -1 \end{cases}$$

We have that, conditioned on $X_i$, $(A_i^+, A_i^-) \overset{d}{=} (A_i, A_i')$. Then

$$\mathbb{E}_{\mathbb{P}_\beta}\left[\exp\left(\lambda \sup_{\zeta \in \mathcal{G}(m)} \frac{1}{n}\sum_{i=1}^n \left(w(A_i, X_i)\zeta(X_i, A_i) - w(A_i', X_i)\zeta(X_i, A_i')\right)\right)\right]$$

$$= \mathbb{E}_{\mathbb{P}_\beta}\left[\exp\left(\lambda \sup_{\zeta \in \mathcal{G}(m)} \frac{1}{n}\sum_{i=1}^n \left(w(A_i^+, X_i)\zeta(X_i, A_i^+) - w(A_i^-, X_i)\zeta(X_i, A_i^-)\right)\right)\right]$$

$$= \mathbb{E}_{\mathbb{P}_\beta, \mathfrak{R}}\left[\exp\left(\lambda \sup_{\zeta \in \mathcal{G}(m)} \frac{1}{n}\sum_{i=1}^n \xi_i\left(w(A_i, X_i)\zeta(X_i, A_i) - w(A_i', X_i)\zeta(X_i, A_i')\right)\right)\right]$$

Our last step is to bound the last line of the above display.

$$= \mathbb{E}_{\mathbb{P}_\beta, \mathfrak{R}}\left[\exp\left(\lambda \sup_{\zeta \in \mathcal{G}(m)} \frac{1}{2}\left(\frac{2}{n}\sum_{i=1}^n \xi_i w(A_i, X_i)\zeta(X_i, A_i) - \frac{2}{n}\sum_{i=1}^n \xi_i w(A_i', X_i)\zeta(X_i, A_i')\right)\right)\right]$$

$$\leq \frac{1}{2}\mathbb{E}_{\mathbb{P}_\beta, \mathfrak{R}}\left[\exp\left(\lambda \sup_{\zeta \in \mathcal{G}(m)} \frac{2}{n}\sum_{i=1}^n \xi_i w(A_i, X_i)\zeta(X_i, A_i)\right)\right]$$

$$+ \frac{1}{2}\mathbb{E}_{\mathbb{P}_\beta, \mathfrak{R}}\left[\exp\left(\lambda \sup_{\zeta \in \mathcal{G}(m)} \frac{2}{n}\sum_{i=1}^n (-\xi_i) w(A_i', X_i)\zeta(X_i, A_i')\right)\right]$$

$$= \mathbb{E}_{\mathbb{P}_\beta, \mathfrak{R}}\left[\exp\left(\lambda \sup_{\zeta \in \mathcal{G}(m)} \frac{2}{n}\sum_{i=1}^n \xi_i w(A_i, X_i)\zeta(X_i, A_i)\right)\right]$$

$\square$

**Lemma C.4.** *Let $G(t; X, \pi) = \mathbb{E}_\mathbb{P}[\mathbb{1}_{\{R \leq t\}}|X]$ be the conditional CDF of returns for all $x \in \mathcal{X}$. Then for $\delta \in (0, 1]$,*

$$\mathbb{P}_\beta\left(\sup_t \left|\frac{1}{n}\sum_{i=1}^n G(t; X_i, \pi) - F(t)\right| \geq \sqrt{\frac{32}{n}\log\frac{(2n)^{1/2}}{\delta}}\right) \leq \delta.$$

*Proof.* Since $G$ is a valid CDF, we apply Lemma C.2 to $\overline{G}$. Consider a function of the form

$$\overline{\zeta}(t; s^1, ..., s^m) = \frac{1}{m}\sum_{j=1}^m \mathbb{1}_{\{s^i \leq t\}}$$

The function $\overline{\zeta}$ can be seen as a stepwise CDF function, where each step is $1/m$ and occurs at points $\{s^j\}_{j=1}^m$.

Lemma C.2 approximates $\overline{G}$ using such $1/m$-stepwise CDFs. For each context $x$, let $s_x^1, ..., s_x^m \in \mathbb{Q}^m$ be the points chosen according to the deterministic procedure in Lemma C.2, such that the following inequality holds:

$$\sup_t \left|G(t; x, \pi) - \overline{\zeta}\left(t; \{s_x^j\}_{j=1}^m\right)\right| \leq \frac{1}{2m}. \tag{18}$$

Next, consider the class of functions

$$\mathcal{G}(m) := \left\{\zeta(s^1, ..., s^m) := \frac{1}{m}\varrho\sum_{j=1}^m \mathbb{1}_{\{s^j \leq t\}} : \forall t \in \mathbb{R}, \varrho \in \{-1, +1\}; \{s^j\}_{j=1}^m \in \mathbb{Q}^m\right\}$$

Note that, $\overline{\zeta}$ is a subset of the function class $\mathcal{G}(m)$, e.g. $\overline{\zeta}\left(t; \{s_x^j\}_{j=1}^m\right) \in \mathcal{G}(m)$.

Then our problem becomes

$$\sup_t \left| \frac{1}{n} \sum_{i=1}^{n} G(t; X_i, \pi) - F(t) \right|$$

$$= \sup_t \left| \frac{1}{n} \sum_{i=1}^{n} G(t; X_i, \pi) - \mathbb{E}_{\mathbb{P}_\beta} \left[ \frac{1}{n} \sum_{i=1}^{n} G(t; X, \pi) \right] \right|$$

$$\leq \sup_t \left| \frac{1}{n} \sum_{i=1}^{n} \overline{\zeta} \left( t; \{s_{X_i}^j\}_{j=1}^m \right) - \mathbb{E}_{\mathbb{P}_\beta} \left[ \frac{1}{n} \sum_{i=1}^{n} \overline{\zeta} \left( t; \{s_{X_i}^j\}_{j=1}^m \right) \right] \right| + \frac{1}{m}$$

$$\leq \sup_{\zeta \in \mathcal{G}(m)} \left| \frac{1}{n} \sum_{i=1}^{n} \zeta \left( \{s_{X_i}^j\}_{j=1}^m \right) - \mathbb{E}_{\mathbb{P}_\beta} \left[ \frac{1}{n} \sum_{i=1}^{n} \zeta \left( \{s_{X_i}^j\}_{j=1}^m \right) \right] \right| + \frac{1}{m}$$

We can now upper bound the RHS. Going forward, we refer to $\zeta(\{s_X^j\}_{j=1}^m)$ as $\zeta(X)$ for short. Then for $\lambda > 0$ we have:

$$\mathbb{E}_{\mathbb{P}_\beta} \left[ \exp \left( \lambda \sup_{\zeta \in \mathcal{G}(m)} \left( \frac{1}{n} \sum_{i=1}^{n} \zeta(X_i) - \mathbb{E}_{\mathbb{P}_\beta} \left[ \frac{1}{n} \sum_{i=1}^{n} \zeta(X_i) \right] \right) \right) \right]$$

$$= \mathbb{E}_{\mathbb{P}_\beta} \left[ \exp \left( \lambda \sup_{\zeta \in \mathcal{G}(m)} \left( \frac{1}{n} \sum_{i=1}^{n} \mathbb{E}_{\mathbb{P}_\beta} \left[ \zeta(X_i) - \zeta(X_i') \Big| \{X_i\}_{i=1}^n \right] \right) \right) \right]$$

$$\leq \mathbb{E}_{\mathbb{P}_\beta} \left[ \exp \left( \lambda \mathbb{E}_{\mathbb{P}_\beta} \left[ \sup_{\zeta \in \mathcal{G}(m)} \frac{1}{n} \sum_{i=1}^{n} (\zeta(X_i) - \zeta(X_i')) \Big| \{X_i\}_{i=1}^n \right] \right) \right]$$

$$\leq \mathbb{E}_{\mathbb{P}_\beta} \left[ \exp \left( \lambda \sup_{\zeta \in \mathcal{G}(m)} \frac{1}{n} \sum_{i=1}^{n} (\zeta(X_i) - \zeta(X_i', A_i')) \right) \right]$$

$$= \mathbb{E}_{\mathbb{P}_\beta, \mathfrak{R}} \left[ \exp \left( \lambda \sup_{\zeta \in \mathcal{G}(m)} \frac{1}{n} \sum_{i=1}^{n} \xi_i (\zeta(X_i) - \zeta(X_i')) \right) \right]$$

$$\leq \mathbb{E}_{\mathbb{P}_\beta, \mathfrak{R}} \left[ \exp \left( 2\lambda \sup_{\zeta \in \mathcal{G}(m)} \frac{1}{n} \sum_{i=1}^{n} \xi_i \zeta(X_i) \right) \right]$$

$$= \mathbb{E}_{\mathbb{P}_\beta, \mathfrak{R}} \left[ \sup_{\zeta \in \mathcal{G}(m)} \exp \left( 2\lambda \frac{1}{n} \sum_{i=1}^{n} \xi_i \zeta(X_i) \right) \right]$$

$$= \mathbb{E}_{\mathbb{P}_\beta, \mathfrak{R}} \left[ \sup_{t, \varrho} \exp \left( 2\lambda \frac{\varrho}{nm} \sum_{j=1}^{m} \sum_{i=1}^{n} \xi_i \mathbb{1}_{\{s_{X_i}^j \leq t\}} \right) \right] \tag{19}$$

where $\{X'\}_i^n$ are the ghost variables, and the last line uses the definition of $\zeta(X_i) = \zeta(s_{X_i}^1, ..., s_{X_i}^m)$.

Now, for each $j$, permute the indices $i$ such that $s_{X_{j(1)}}^j \leq ... \leq s_{X_{j(i)}}^j \leq ... \leq s_{X_{j(n)}}^j$. Then, for a given $j$, consider the function

$$\sum_{i=1}^{n} \xi_{j(i)} \mathbb{1}_{\{s_{X_{j(i)}}^j \leq t\}},$$

which equals

1. $0$ if $t < s_{X_{j(1)}}^j$,

2. $\varrho \sum_{i=1}^{k} \xi_{j(i)}$ if there exists $k \in \{1, ..., n-1\}$ such that $s_{X_{j(k)}}^j \leq t \leq s_{X_{j(k)+1}}^j$,

3. $\varrho \sum_{i=1}^{n} \xi_{j(i)}$ otherwise.

Then the RHS of (19) equals

$$\mathbb{E}_{\mathbb{P}_\beta,\mathfrak{R}} \left[ \sup_{t,\varrho} \exp \left( 2\lambda \frac{\varrho}{nm} \sum_{j=1}^m \sum_{i=1}^n \xi_i \mathbb{1}_{\{s^j_{X_i} \leq t\}} \right) \right]$$

$$= \mathbb{E}_{\mathbb{P}_\beta,\mathfrak{R}} \left[ \max_{k,\varrho} \exp \left( 2\lambda \frac{\varrho}{nm} \sum_{j=1}^m \sum_{i=1}^k \xi_{j(i)} \right) \right]$$

$$\leq \mathbb{E}_{\mathbb{P}_\beta,\mathfrak{R}} \left[ \max_{j,k,\varrho} \exp \left( 2\lambda \frac{\varrho}{n} \sum_{i=1}^k \xi_{j(i)} \right) \right].$$

Further, we have that

$$\max_{j,k,\varrho} \exp \left( 2\lambda \frac{\varrho}{n} \sum_{i=1}^k \xi_{j(i)} \right)$$

$$= \max_{j,k} \left( \exp \left( \frac{2\lambda}{n} \sum_i^k \xi_{j(i)} \right) \mathbb{1}_{\{\sum_i^k \xi_{j(i)} \geq 0\}} + \exp \left( -\frac{2\lambda}{n} \sum_i^k \xi_{j(i)} \right) \mathbb{1}_{\{\sum_i^k \xi_{j(i)} < 0\}} \right)$$

$$\leq 2 \max_{j,k} \exp \left( \frac{2\lambda}{n} \sum_i^k \xi_{j(i)} \right) \mathbb{1}_{\{\sum_i^k \xi_{j(i)} \geq 0\}}.$$

Putting it together, we have that

$$\mathbb{E}_{\mathbb{P}_\beta} \left[ \exp \left( \lambda \sup_{\zeta \in \mathcal{G}(m)} \left| \frac{1}{n} \sum_{i=1}^n \zeta(X_i) - \mathbb{E}_{\mathbb{P}_\beta} \left[ \frac{1}{n} \sum_{i=1}^n \zeta(X_i) \right] \right| \right) \right]$$

$$\leq 2\mathbb{E}_{\mathbb{P}_\beta,\mathfrak{R}} \left[ \max_{j,k} \exp \left( \frac{2\lambda}{n} \sum_i^k \xi_{j(i)} \right) \mathbb{1}_{\{\sum_i^k \xi_{j(i)} \geq 0\}} \right] \qquad (20)$$

Now we are left to bound the RHS of (20). Using Lemma B.1,

$$\mathbb{E}_{\mathbb{P}_\beta,\mathfrak{R}} \left[ \max_{j,k} \exp \left( \frac{2\lambda}{n} \sum_i^k \xi_{j(i)} \right) \mathbb{1}_{\{\max_k \sum_i^k \xi_{j(i)} \geq 0\}} \right]$$

$$\leq \mathbb{P}_\beta \left( \max_k \frac{2\lambda}{n} \sum_i^k \xi_{j(i)} \geq 0 \right) + 2\lambda \sum_j \int_0^\infty \exp(\lambda t) \mathbb{P} \left( \frac{2\lambda}{n} \sum_{i=1}^n \xi_{j(i)} \geq t \right) dt.$$

Similarly, for any $j$, we have

$$\mathbb{E}_{\mathbb{P}_\beta,\mathfrak{R}} \left[ \exp \left( \frac{2\lambda}{n} \sum_i^n \xi_{j(i)} \right) \mathbb{1}_{\{\sum_i^k \xi_{j(i)} \geq 0\}} \right]$$

$$= \mathbb{P}_\beta \left( \frac{2\lambda}{n} \sum_i^n w(A_{j(i)}, X_{j(i)}) \xi_{j(i)} \geq 0 \right) + \lambda \int_0^\infty \exp(\lambda t) \mathbb{P} \left( \frac{2\lambda}{n} \sum_{i=1}^n \xi_{j(i)} \geq t \right) dt$$

Putting these two together, we have

$$\mathbb{E}_{\mathbb{P}_\beta,\mathfrak{R}} \left[ \max_{j,k} \exp \left( \frac{2\lambda}{n} \sum_i^k \xi_{j(i)} \right) \mathbb{1}_{\{\sum_i^k \xi_{j(i)} \geq 0\}} \right]$$

$$\leq \sum_j \mathbb{P}_\beta \left( \max_k \frac{2\lambda}{n} \sum_i^k \xi_{j(i)} \geq 0 \right) - 2 \sum_j \mathbb{P}_\beta \left( \frac{2\lambda}{n} \sum_i^n \xi_{j(i)} \geq 0 \right)$$

$$+ 2 \sum_j \mathbb{E}_{\mathbb{P}_\beta,\mathfrak{R}} \left[ \exp \left( \frac{2\lambda}{n} \sum_i^n \xi_{j(i)} \right) \mathbb{1}_{\{\sum_i^n \xi_{j(i)} \geq 0\}} \right]$$

$$\leq 2 \sum_j \mathbb{E}_{\mathbb{P}_\beta, \mathfrak{R}} \left[ \exp\left( \frac{2\lambda}{n} \sum_i^n \xi_{j(i)} \right) \mathbb{1}_{\{\sum_i^n \xi_{j(i)} \geq 0\}} \right]$$

$$\leq 2m \mathbb{E}_{\mathbb{P}_\beta, \mathfrak{R}} \left[ \exp\left( \frac{2\lambda}{n} \sum_i^n \xi_{j(i)} \right) \right]$$

$$\leq 2m \exp\left( \frac{2\lambda^2}{n} \right)$$

where the last inequality uses the fact that $\xi$ is a Rademacher random variable. Finally, using Markov's inequality,

$$\mathbb{P}_\beta \left( \sup_t \left| \frac{1}{n} \sum_{i=1}^n G(t; X_i, \pi) - F(t) \right| \geq \epsilon + \frac{1}{m} \right)$$

$$\leq \mathbb{P}_\beta \left( \exp\left( \lambda \sup_t \left| \frac{1}{n} \sum_{i=1}^n \zeta(X_i) - \mathbb{E}_{\mathbb{P}_\beta} \left[ \frac{1}{n} \sum_{i=1}^n \zeta(X_i) \right] \right| \right) \geq \exp(\lambda\epsilon) \right)$$

$$\leq 4m \exp\left( \frac{2\lambda^2}{n} - \lambda\epsilon \right)$$

Because this holds for any $\lambda > 0$, we can minimize the RHS over $\lambda$:

$$\mathbb{P}_\beta \left( \sup_t \left| \frac{1}{n} \sum_{i=1}^n G(t; X_i, \pi) - F(t) \right| \geq \epsilon + \frac{1}{m} \right) \leq \inf_{\lambda > 0} 4m \exp\left( \frac{2\lambda^2}{n} - \lambda\epsilon \right)$$

$$= 4m \exp\left( \frac{-n\epsilon}{8} \right).$$

Then we have

$$\mathbb{P}_\beta \left( \sup_t \left| \frac{1}{n} \sum_{i=1}^n G(t; X_i, \pi) - F(t) \right| \geq \sqrt{\frac{8}{n} \log \frac{4m}{\delta}} + \frac{1}{m} \right) \leq \delta.$$

Setting $m = \sqrt{n/8}$ gives the theorem statement:

$$\mathbb{P}_\beta \left( \sup_t \left| \frac{1}{n} \sum_{i=1}^n G(t; X_i, \pi) - F(t) \right| \geq \sqrt{\frac{32}{n} \log \frac{(2n)^{1/2}}{\delta}} \right) \leq \delta.$$

$\square$

## D  Proofs for Risk Functional Estimation (Section 6)

*Proof of Theorem 6.1.* By the definition of $L$-Lipschitz risk functionals, for the CDFs $F$ and $\widehat{F}$,

$$|\rho(\widehat{F}) - \rho(F)| \leq L\|\widehat{F} - F\|_\infty$$
$$\leq L\epsilon$$

with probability at least $1 - \delta$, where the last line uses the fact that $\widehat{F}$ is $\epsilon$-close to $F$ with probability at least $1 - \delta$. $\square$

# E   Risk Estimation with Unknown Behavior Policy

We begin this section with a consideration of estimators when the behavior policy is unknown, and must be modeled or estimated, which we call $\widehat{\beta}$. We first define the IS, DR, and DI estimators using $\widehat{\beta}$, then derive their bias and variance expressions. To differentiate between the estimator that use $\beta$ and the estimators that use $\widehat{\beta}$, we call the latter $\widetilde{F}$ while continuing to call the former $\widehat{F}$.

The proofs of bias and variance begins with derivations for the DR estimator with estimated policy, from which the bias and variance of the remaining estimators can be derived as special cases.

Let $\widehat{\beta}$ be the estimated behavior policy, and let $\widehat{w}(a, x) := \frac{\pi(a|x)}{\widehat{\beta}(a|x)}$ be the importance weight with estimated policy. Then the importance sampling (IS) estimator is given by

$$\widetilde{F}_{\text{IS}}(t) := \frac{1}{n} \sum_{i=1}^{n} \widehat{w}(a_i, x_i) \mathbb{1}_{\{r_i \leq t\}} \tag{21}$$

Then doubly robust (DR) estimator is:

$$\widetilde{F}_{\text{DR}}(t) := \frac{1}{n} \sum_{i=1}^{n} \widehat{w}(a_i, x_i) \left( \mathbb{1}_{\{r_i \leq t\}} - \overline{G}(t; x_i, a_i) \right) + \overline{G}(t; x_i, \pi) \tag{22}$$

And the direct method (DI) estimator is still defined to be

$$\widehat{F}_{\text{DI}}(t) := \frac{1}{n} \sum_{i=1}^{n} \overline{G}(t; x_i, \pi) \tag{23}$$

Note that the direct estimator does not depend on the behavior policy, and thus we continue to call it $\widetilde{F}_{\text{DI}}$.

## E.1   Bias and Variance

Next, we analyze the bias and variance of these estimators. Define $\Delta(a, x, t)$ to be the additive error between $G$ and the model $\overline{G}$, and define $\delta(x, a)$ to be the multiplicative error of the estimate $\widehat{\beta}$, that is:

$$\Delta(t; x, a) := \overline{G}(t; x, a) - G(t; x, a),$$
$$\delta(x, a) := 1 - \beta(a|x)/\widehat{\beta}(a|x).$$

Note that when $\beta$ is known or $\widehat{\beta} = \beta$ for all $x, a$, $\delta(x, a) = 0$, The bias of the IS estimator then given in Lemma E.1, in terms of $\delta$ and the conditional reward distribution $G$.

**Lemma E.1** (Bias and Variance of IS Estimator with $\widehat{\beta}$.). *The expectation of the IS estimator is*

$$\mathbb{E}_{\mathbb{P}_\beta}[\widetilde{F}_{IS}(t)] = F(t) + \mathbb{E}_{\mathbb{P}}[\delta(A, X)G(t; X, \pi)]$$

*When $\widehat{\beta}(a|x) = \beta(a|x)$ for all $a, x$, the IS estimator is unbiased and $\mathbb{E}_{\mathbb{P}_\beta}[\widehat{F}_{IS}(t)] = F(t)$. Further, the variance is*

$$\mathbb{V}_{\mathbb{P}_\beta}[\widetilde{F}_{IS}(t)] = \frac{1}{n} \mathbb{E}_{\mathbb{P}} \left[ (1 - \delta(A, X))^2 \sigma^2(t; X, A) \right] + \frac{1}{n} \mathbb{V}_{\mathbb{P}} \left[ \mathbb{E}_{\mathbb{P}} \left[ (1 - \delta(A, X))G(t; X, A)|X \right] \right]$$
$$+ \frac{1}{n} \mathbb{E}_{\mathbb{P}} \left[ \mathbb{V}_{\mathbb{P}_\beta} \left[ \widehat{w}(A, X)G(t; X, A)|X \right] \right] \tag{24}$$

The expression for variance is broken down into three terms. The first represents randomness in the rewards, and the second represents variance from the aleatoric uncertainty due to randomness over contexts $X$. The final term represents variance arising from using importance sampling, and is proportional to the true CDF of conditional rewards $G$.

The following lemma, similarly, derives the bias and variance for the DR estimator:

**Lemma E.2** (Bias and Variance of DR Estimator with $\widehat{\beta}$.)**.** *The pointwise expectation of the DR estimator is*

$$\mathbb{E}_{\mathbb{P}_\beta}[\widetilde{F}_{DR}(t)] = F(t) + \mathbb{E}_{\mathbb{P}}[\delta(X, A)\Delta(t; X, A)]$$

*Further, when there is perfect knowledge of the behavior policy $\beta$, e.g. $\hat{\beta}(a|x) = \beta(a|x)$ for all $a, x$, the DR estimator is unbiased and*

$$\mathbb{E}_{\mathbb{P}_\beta}[\widetilde{F}_{DR}(t)] = F(t)$$

*The variance of the doubly robust estimator is given by*

$$\mathbb{V}_{\mathbb{P}_\beta}[\widetilde{F}_{DR}(t)] = \frac{1}{n}\mathbb{E}_{\mathbb{P}}\left[(1 - \delta(A, X))^2 \sigma^2(t; X, A)\right] + \frac{1}{n}\mathbb{V}_{\mathbb{P}}\left[\mathbb{E}_{\mathbb{P}}\left[\delta(A, X)\Delta(t; X, A) + G(t; X, A)|X\right]\right]$$

$$+ \frac{1}{n}\mathbb{E}_{\mathbb{P}}\left[\mathbb{V}_{\mathbb{P}_\beta}\left[\widehat{w}(A, X)\Delta(t; X, A)|X\right]\right] \tag{25}$$

Because the DR estimator takes advantage of both policy and reward estimates, it is unbiased whenever either the estimated policy or estimated reward is unbiased. Further, when we have access to the true behavior policy $\beta$ and $\widehat{w} = w$, it retains the unbiasedness of the IS estimator.

Compared to the IS estimator, the DR estimator may also have pointwise reduced variance. When the variances of the IS estimator (24) and the DR estimator (25) are compared, the first term is identical, and the middle term is of similar magnitude because the randomness in contexts $X$ is endemic. The third term is the primary difference. For the IS estimator, it is proportional to $G$, but for the DR estimator, it is proportional to the error $\Delta$ between the estimated conditional CDF $\overline{G}$ and the true $G$. Thus, this term can be much larger in the IS estimator when $\widehat{w}$ is large and the error $\Delta$ is smaller than $G$. This demonstrates that the DR estimator retains the low bias of the IS estimator, but has the advantage of reduced variance.

Next, Lemma E.3 gives the bias and variance of the DI estimator, which is directly related to the bias and variance of the conditional distribution model $\overline{G}$.

**Lemma E.3** (Bias of DI Estimator with $\widehat{\beta}$.)**.** *The bias is*

$$\mathbb{E}_{x,a\sim\beta,r}[\widetilde{F}_{DI}(t)] = F(t) + \mathbb{E}_{\mathbb{P}}[\Delta]$$

*and the variance is*

$$\mathbb{V}[\widetilde{F}_{DI}(t)] = \frac{1}{n}\mathbb{V}_{\mathbb{P}}\left[G(t; X, \pi) + \Delta\right].$$

While the DI estimator has lower variance than both the IS and DR estimators, it suffers from potentially high bias from $\overline{G}$. Unlike the other two estimators, it is biased even when $\widehat{\beta}$ is a perfect estimate of $\beta$, which in practice is undesirable. Though the DI estimator has low bias when $\overline{G}$ is a good model of the condition reward distribution, it is often much easier to form accurate models of $\beta$ than of $G$.

**Proofs: Bias and Variance**

We begin by proving the bias and variance expressions of the DR estimator with $\widehat{\beta}$. The bias and variance of the other estimators can be derived as special cases, which we show later.

*Proof of Lemma E.2.* First, we take the expectation of the DR estimator (22) with respect to $\mathbb{P}_\beta$:

$$\mathbb{E}_{\mathbb{P}_\beta}\left[\widetilde{F}(t)\right] = \mathbb{E}_{\mathbb{P}_\beta}\left[\frac{\pi(A|X)}{\widehat{\beta}(A|X)}\mathbb{1}\{R \le t\}\right] + \mathbb{E}_{\mathbb{P}_\beta}\left[\left(\frac{\pi(A|X)}{\widehat{\beta}(A|X)} - \sum_a \pi(A|X)\right)\overline{G}(t; X, A)\right]$$

$$= \mathbb{E}_{\mathbb{P}}\left[\frac{\beta(A|X)}{\widehat{\beta}(A|X)}\mathbb{1}\{R \le t\}\right] + \mathbb{E}_{\mathbb{P}}\left[\left(\frac{\beta(A|X)}{\widehat{\beta}(A|X)} - 1\right)\overline{G}(t; X, A)\right]$$

$$= F(t) + \mathbb{E}_{\mathbb{P}}\left[\left(\frac{\beta(A|X)}{\widehat{\beta}(A|X)} - 1\right)\mathbb{1}\{R \le t\}\right] + \mathbb{E}_{\mathbb{P}}\left[\left(\frac{\beta(A|X)}{\widehat{\beta}(A|X)} - 1\right)\overline{G}(t; X, A)\right]$$

$$= F(t) + \mathbb{E}_{\mathbb{P}}\left[\left(\frac{\beta(A|X)}{\widehat{\beta}(A|X)} - 1\right)\left(G(t; X, A) - \overline{G}(t; X, A)\right)\right]$$

$$= F(t) + \mathbb{E}_{\mathbb{P}}\left[\delta(A, X)\Delta(t; X, A)\right]$$

When $\widehat{\beta} = \beta$ for all $a, x$, we have $\delta = 0$, giving the unbiasedness of the estimator.

Starting from the second line of the proof of variance for the DR estimator (Appendix C.1), we have

$$
\begin{aligned}
\mathbb{V}_{\mathbb{P}_\beta}\left[\widetilde{F}_{\text{DR}}(t)\right] &= \frac{1}{n}\mathbb{V}_{\mathbb{P}_\beta}\left[\widehat{w}(A, X)\left(\mathbb{1}_{\{R \le t\}} - \overline{G}(t; X, A)\right) + \overline{G}(t; X, \pi)\right] \\
&= \frac{1}{n}\mathbb{E}_{\mathbb{P}_\beta}\left[\widehat{w}(A, X)^2\sigma^2(t; X, A)\right] \\
&\quad + \frac{1}{n}\mathbb{V}_{\mathbb{P}_\beta}\left[\widehat{w}(A, X)\left(G(t; X, A) - \overline{G}(t; X, A)\right) + \overline{G}(t; X, \pi)\right] \\
&= \frac{1}{n}\mathbb{E}_{\mathbb{P}}\left[\left(\frac{\beta(A, X)}{\widehat{\beta}(A, X)}\right)^2\sigma^2(t; X, A)\right] \\
&\quad + \frac{1}{n}\mathbb{V}_{\mathbb{P}_\beta}\left[\mathbb{E}_{\mathbb{P}_\beta}\left[\widehat{w}(A, X)\left(G(t; X, A) - \overline{G}(t; X, A)\right) + \overline{G}(t; X, \pi)|X\right]\right] \\
&\quad + \frac{1}{n}\mathbb{E}_{\mathbb{P}_\beta}\left[\mathbb{V}_{\mathbb{P}_\beta}\left[\widehat{w}(A, X)\left(G(t; X, A) - \overline{G}(t; X, A)\right)|X\right]\right] \\
&= \frac{1}{n}\mathbb{E}_{\mathbb{P}}\left[\left(1 - \delta(A, X)\right)^2\sigma^2(t; X, A)\right] \\
&\quad + \frac{1}{n}\mathbb{V}_{\mathbb{P}}\left[\mathbb{E}_{\mathbb{P}}\left[\delta(A, X)\Delta(t; X, A) + G(t; X, A)|X\right]\right] \\
&\quad + \frac{1}{n}\mathbb{E}_{\mathbb{P}}\left[\mathbb{V}_{\mathbb{P}_\beta}\left[\widehat{w}(A, X)\Delta(t; X, A)|X\right]\right]
\end{aligned}
$$

the second line uses a change of measure in the first term, and the law of total variance conditioned on the context $X$. The third line follows again from change of measure and substituting in the definition of $\delta$ and $\Delta$.

$\square$

Lemma E.1 is derived from Lemma E.2 using the fact that the IS estimator is a special case of the DR estimator with $\overline{G} = 0$.

Lemma E.3 is derived from Lemma E.2 by using $\widehat{\beta} \to \infty$ which means $\widehat{w} = 0$, e.g. importance weighting is not used, and $\delta = 1$.

## E.2 CDF and Risk Estimate Error Bounds

Theorem E.1 generalizes the CDF error bounds established for the IS and DR estimators with known behavior policy to the case where $\widehat{\beta}$ is estimated, given an additional high-probability guarantee on the quality of $\widehat{\beta}$.

**Theorem E.1.** *For the IS or DR CDF estimator $\widetilde{F}$ that uses estimated weights $\widehat{w}(a, x) = \pi(a|x)/\widehat{\beta}(a, x)$, given an estimate $\widehat{\beta}$ that is $\epsilon_\beta$-close to the true behavior policy $\beta$, that is*

$$
\sup_{a, x}|\beta(a|x) - \widehat{\beta}(a|x)| \le \epsilon_\beta,
$$

*we have with probability at least $1 - \delta$ that*

$$
\mathbb{P}_\beta\left(\sup_{t \in \mathbb{R}}\left|\widetilde{F}(t) - F(t)\right| \le \epsilon + c\epsilon_\beta\right) \ge 1 - \delta
$$

*where $\epsilon$ is either $\epsilon_{IS}$ or $\epsilon = \epsilon_{DR}$ depending the choice of $\widehat{F}$, and $c = w_{max}\left(\inf_{a, x}\widehat{\beta}(a|x)\right)^{-1}$.*

Similarly, for $L$-Lipschitz risk functionals, the general error bound given in Theorem 6.1 can be extended to the case of $\widehat{\beta}$ by adding the additional error term from the policy estimation.

**Corollary E.1.** *For the IS or DR CDF estimator $\widetilde{F}$ that uses estimated weights $\widehat{w}(a,x) = \pi(a|x)/\widehat{\beta}(a,x)$, given an estimate $\widehat{\beta}$ that is $\epsilon_\beta$-close to the true behavior policy $\beta$, we have with probability at least $1 - \delta$ that*

$$\left| \rho(\widetilde{F}) - \rho(F) \right| \leq L \left( \epsilon + c\epsilon_\beta \right)$$

*where $c = w_{max} \left( \inf_{a,x} \widehat{\beta}(a|x) \right)^{-1}$.*

Note that the error contributed by policy estimation, $c\epsilon_\beta$, is primarily dependent upon two factors. First, the quality of $\widehat{\beta}$ estimation determines the magnitude of $\epsilon_\beta$; a poor estimate naturally leads to a higher value of this constant. Second, $c$ is a problem-dependent constant proportional to the maximum importance weight $w_{max}$ and the minimum probability of the estimated behavior policy $\inf_{a,x} \widehat{\beta}(a|x)$. If $\inf_{a,x} \widehat{\beta}(a|x)$ is particularly small, the error bound is also large. This reflects the fact that CDF estimation can be difficult when the behavior policy places low probability in some area of the context and action space.

**Remark E.1.** *When actions and contexts are discrete, and $\widehat{\beta}$ is estimated using empirical averages, standard concentrations for the mean of a random variable can be used to determine $\epsilon_\beta$. If $\widehat{\beta}$ is estimated using regression, depending on the estimator $\epsilon_\beta$ can also be determined from concentration inequalities.*

v2

**Proofs: Error Bounds**

The proof of these results is given below.

*Proof of Theorem E.1.* We can decompose the error $\widehat{F} - F$ as:

$$\sup_t |\widetilde{F}(t) - F(t)| \leq \sup_t \left( |\widehat{F}(t) - F(t)| + |\widetilde{F}(t) - \widehat{F}(t)| \right)$$
$$\leq \sup_t |\widehat{F}(t) - F(t)| + \sup_t |\widetilde{F}(t) - \widehat{F}(t)|$$

Theorem 5.1 gives a bound for the first term, and the bound for the second term bound is given in Lemma E.4 for the IS estimator, and in Lemma E.5 for the DR estimator. $\square$

*Proof of Corollary E.1.* This result follows directly from applying the general risk estimation error bound in Theorem 6.1 to the error from Theorem E.1. $\square$

The intermediary lemmas are defined and proved below:

**Lemma E.4.** *Suppose that $|\widehat{\beta}(a|x) - \beta(a|x)| \leq \epsilon_\beta$ for all $a, x$ with probability at least $1 - \delta$. Then with probability at least $1 - \delta$,*

$$\sup_t |\widetilde{F}_{IS}(t) - \widehat{F}_{IS}(t)| \leq c\epsilon_\beta$$

*where $c = w_{max} \left( \inf_{a,x} \widehat{\beta}(a|x) \right)^{-1}$.*

*Proof.* We can bound the LHS of the lemma statement as follows.

$$\sup_t |\widetilde{F}_{\text{IS}}(t) - \widehat{F}_{\text{IS}}(t)| = \sup_t \left| \frac{1}{n} \sum_{i=1}^n (w(a_i, x_i) - \widehat{w}(a_i, x_i)) \mathbb{1}_{\{r_i \le t\}} \right|$$

$$= \sup_t \left| \frac{1}{n} \sum_{i=1}^n \left( \frac{\pi(a_i|x_i)}{\beta(a_i|x_i)} - \frac{\pi(a_i|x_i)}{\widehat{\beta}(a_i|x_i)} \right) \mathbb{1}_{\{r_i \le t\}} \right|$$

$$\le \frac{1}{n} \sum_{i=1}^n \left| \frac{\pi(a_i|x_i)}{\beta(a_i|x_i)} - \frac{\pi(a_i|x_i)}{\widehat{\beta}(a_i|x_i)} \right|$$

$$\le w_{max} \frac{1}{n} \sum_{i=1}^n \left| 1 - \frac{\beta(a_i|x_i)}{\widehat{\beta}(a_i|x_i)} \right|$$

$$= w_{max} \frac{1}{n} \sum_{i=1}^n \left| \frac{\widehat{\beta}(a_i|x_i) - \beta(a_i|x_i)}{\widehat{\beta}(a_i|x_i)} \right|$$

$$\le w_{max} \left( \inf_{a,x} \widehat{\beta}(a|x) \right)^{-1} \frac{1}{n} \sum_{i=1}^n \left| \widehat{\beta}(a_i|x_i) - \beta(a_i|x_i) \right|$$

$$\le w_{max} \left( \inf_{a,x} \widehat{\beta}(a|x) \right)^{-1} \epsilon_\beta$$

where the last line follows from using the assumption that $|\widehat{\beta}(a|x) - \beta(a|x)| \le \epsilon_\beta$ for all $a, x$. □

**Lemma E.5.** *Suppose that $|\widehat{\beta}(a|x) - \beta(a|x)| \le \epsilon_\beta$ for all $a, x$ with probability at least $1 - \delta$. Then with probability at least $1 - \delta$,*

$$\sup_t |\widetilde{F}_{DR}(t) - \widehat{F}_{DR}(t)| \le c\epsilon_\beta$$

*where $c = w_{max} \left( \inf_{a,x} \widehat{\beta}(a|x) \right)^{-1}$.*

*Proof.* We can bound the LHS of the lemma statement as follows. Using the definitions of the DR estimators,

$$\sup_t |\widetilde{F}_{DR}(t) - \widehat{F}_{DR}(t)| = \sup_t \left| \frac{1}{n} \sum_{i=1}^n (w(a_i, x_i) - \widehat{w}(a_i, x_i)) \left( \mathbb{1}_{\{r_i \le t\}} - \overline{G}(t; x_i, a_i) \right) \right|$$

$$= \sup_t \left| \frac{1}{n} \sum_{i=1}^n \left( \frac{\pi(a_i|x_i)}{\beta(a_i|x_i)} - \frac{\pi(a_i|x_i)}{\widehat{\beta}(a_i|x_i)} \right) \left( \mathbb{1}_{\{r_i \le t\}} - \overline{G}(t; x_i, a_i) \right) \right|$$

$$\le \frac{1}{n} \sum_{i=1}^n \left| \frac{\pi(a_i|x_i)}{\beta(a_i|x_i)} - \frac{\pi(a_i|x_i)}{\widehat{\beta}(a_i|x_i)} \right|$$

$$\le w_{max} \frac{1}{n} \sum_{i=1}^n \left| 1 - \frac{\beta(a_i|x_i)}{\widehat{\beta}(a_i|x_i)} \right|$$

$$= w_{max} \frac{1}{n} \sum_{i=1}^n \left| \frac{\widehat{\beta}(a_i|x_i) - \beta(a_i|x_i)}{\widehat{\beta}(a_i|x_i)} \right|$$

$$\le w_{max} \left( \inf_{a,x} \widehat{\beta}(a|x) \right)^{-1} \frac{1}{n} \sum_{i=1}^n \left| \widehat{\beta}(a_i|x_i) - \beta(a_i|x_i) \right|$$

$$\le w_{max} \left( \inf_{a,x} \widehat{\beta}(a|x) \right)^{-1} \epsilon_\beta$$

where the last line uses the assumption that $|\widehat{\beta}(a|x) - \beta(a|x)| \le \epsilon_\beta$ for all $a, x$. □

# F Additional Experiments

**UCI Implementation Details** Following [20, 19, 57], we obtain our off-policy contextual bandit datasets by transforming classification datasets. The contexts are the provided features, and the actions correspond to the possible class labels. To obtain the evaluation policy $\pi$, we use the output probabilities of a trained logistic regression classifier [36]. The behavior policy is defined as $\beta = \alpha\pi + (1 - \alpha)\pi_{\mathrm{UNIF}}$, where $\pi_{\mathrm{UNIF}}$ is a uniform policy over the actions, for some $\alpha \in (0, 1]$. Each dataset is generated by drawing actions for each context according to the probabilities of $\beta$, and the deterministic reward is 1 if the action matches the ground truth label, and 0 otherwise.

We apply this process to the set of 9 UCI datasets [18] used in [20, 19, 57], which each have differing dimensions $d$, actions $k$, and sample size $n$. Models $\overline{G}$ must be constructed for the DM and DR estimators. As in [20], the dataset is divided into two splits, with each of the two splits used to estimate $\overline{G}$, which is then used with the other split to calculate the estimator. The two results are averaged to produce the final estimators. In order to estimate $\overline{G}$, we discretize the reward support into $t \in [0, 1]$, and train a logistic regression classifier [36] for each action $a$ and each $t$, with regularization parameter $C = 1$ and tolerance 0.0001. The code to reproduce these experiments is provided in the supplementary. On a CPU, they take roughly half a day of compute in total.

**Relationship With $\alpha$** We plot the error over the range of $\alpha$, which controls the mismatch between the behavioral policy $\beta$ and the target policy $\pi$ and is thus proportional to $w_{max}$, for the PageBlocks dataset (also in Figure 1). The CDF error is shown in Figure 5 and the mean squared error (MSE) for the mean, CVaR 0.5, and variance risk functionals are shown in Figure 6.

The DR estimator exhibits lower error than any other estimator, and significantly lower variance than the IS and WIS estimators, across the range of $\alpha$. This is particularly obvious in the region where $\alpha$ is small, which is where importance weights can become larger and the IS-based estimators are prone to higher variance. Note that the $\mathrm{CVaR}_{0.5}$ MSE is close to 0 for all estimators.

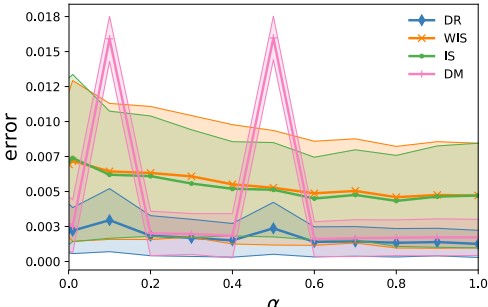

Figure 5: Sup-norm CDF error over $\alpha$ for PageBlocks. Shaded region shows one empirical standard deviation.

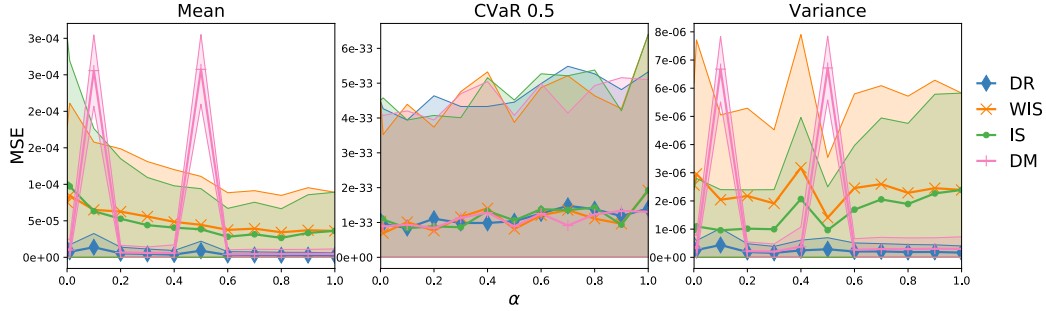

Figure 6: Mean squared error (MSE) over $\alpha$ for different risk functionals evaluated in the PageBlocks dataset. Shaded region shows one empirical standard deviation.

**Evaluation Over UCI Datasets** We display the sup-norm error of the estimated CDF and the mean-squared error (MSE) of estimated risk functionals (mean, CVaR$_{0.5}$, and variance) for the 9 UCI datasets below. Here, $\alpha = 0.5$ is fixed. All plots are shown over 500 repetitions, with error bars omitted for readability but similar to those shown in Figure 1.

The general trends reflect analysis presented in Section 7. As expected of our distribution-based approach, trends in CDF estimation performance are reflected in risk estimation performance. Both the DR and IS estimators exhibit the expected $O(1/\sqrt{n})$ error convergence across the estimation tasks. Generally, the DR estimator does as well as if not better than the other estimators; where the model is difficult to specify well, the DR estimator may suffer slightly in performance in the low sample regime, but always outperforms the other estimators as the number of samples $n$ increases.

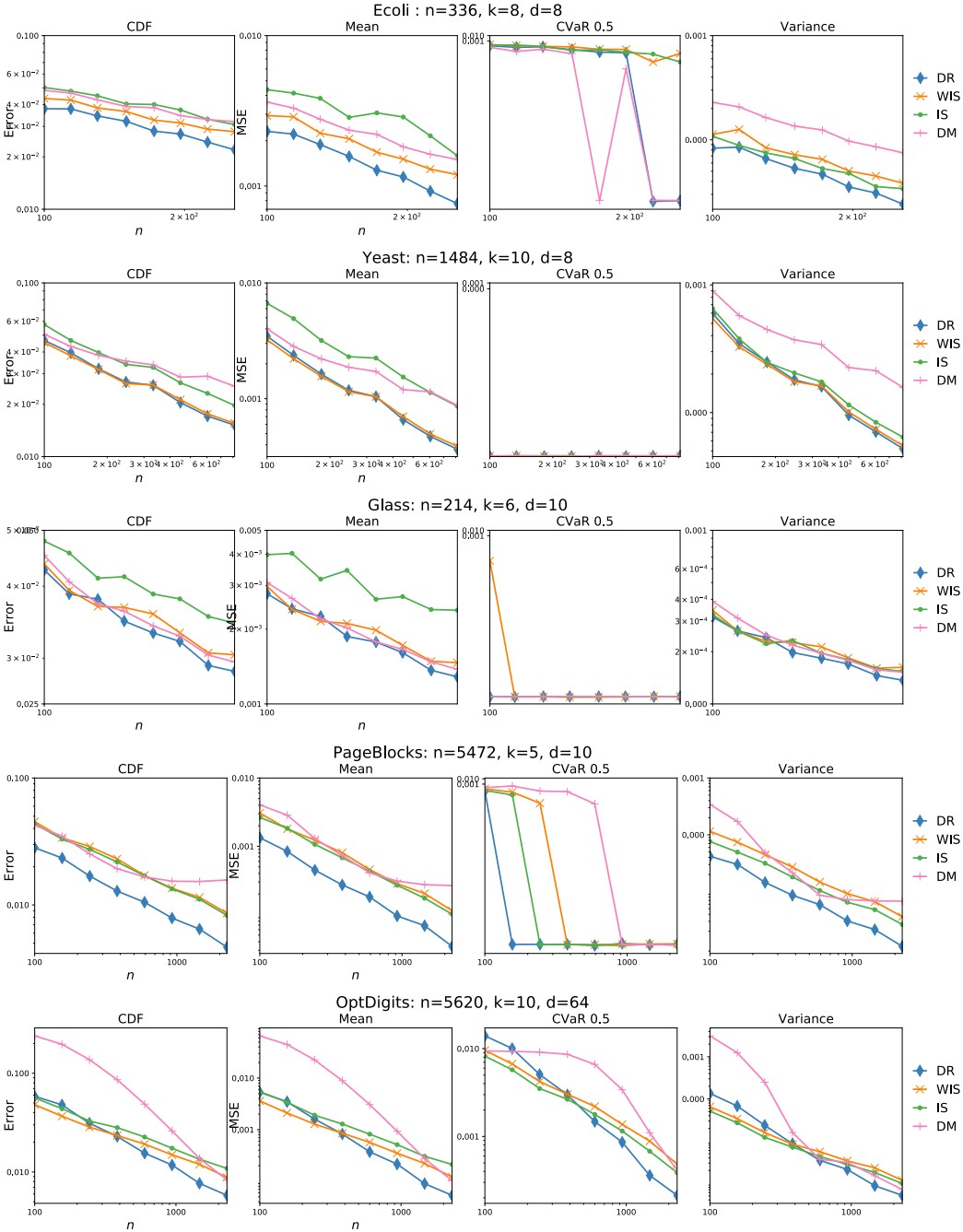

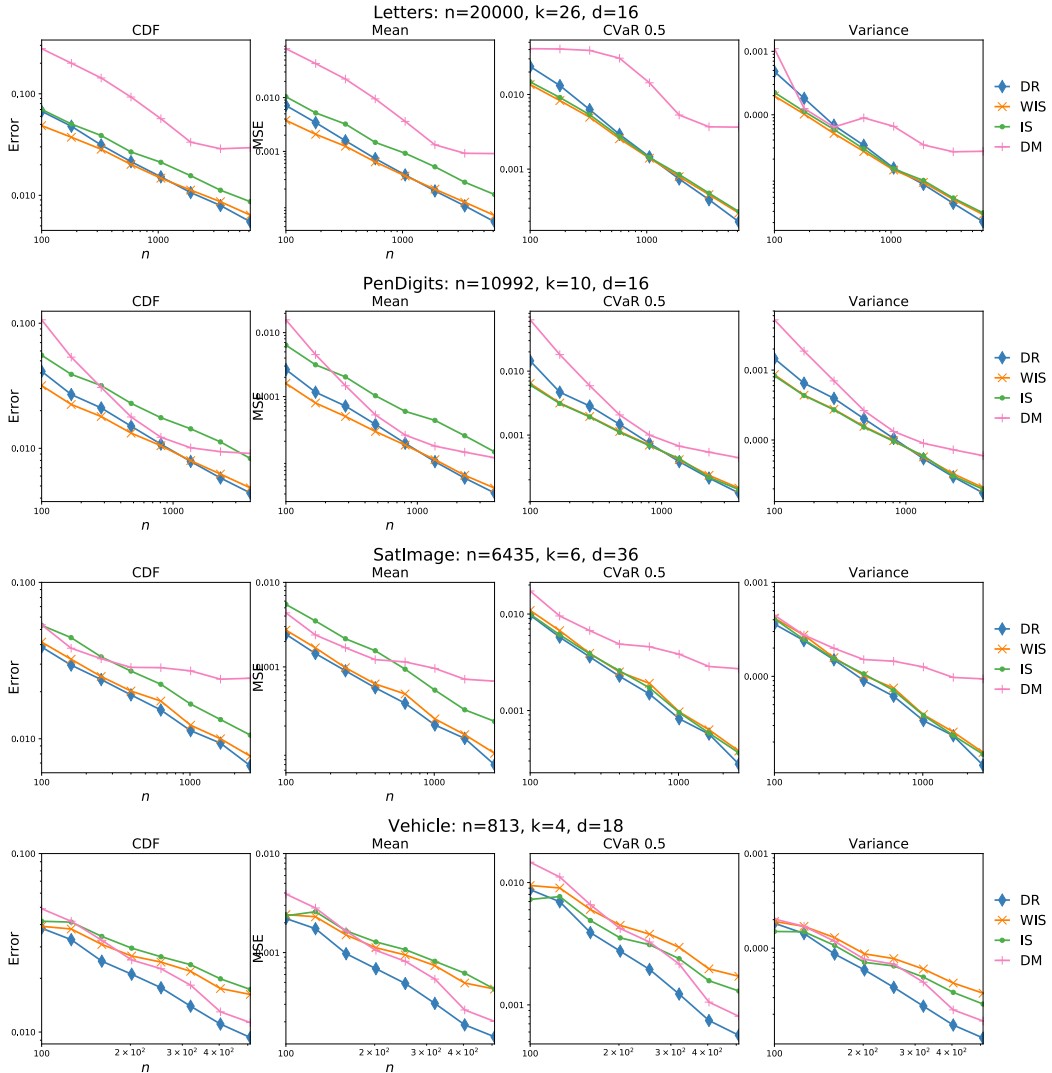