# OpenReview forum: "Off-Policy Risk Assessment in Contextual Bandits"
_NeurIPS.cc/2021/Conference — NeurIPS 2021 Poster_

### Official Review · Reviewer_xxgN · 2021-07-14

**Rating:** 6
**Confidence:** 4

**Summary:**

Previous OPE studies mostly are focused on the overall expected return. This paper studies off-policy evaluation for a more broad class of objectives, i.e., the Lipschitz risk functionals, including CVaR, variance, mean-variance, etc. The main idea is to first use an IPS/DR estimate to estimate the target policy's CDF, then use it as a plug-in estimator to any functionals that satisfy the Lipschitz property. Theoretically, an $O(1/\sqrt{n})$ error bound is derived for the Lipschitz estimate. Empirically, the performance of the plug-in estimator (based on different CDF estimates) is examined.

OPE is crucial to estimate the performance of any new policy while online A/B testing or experimentation might be infeasible. Most prior works on OPE focus on efficient estimation of the expected reward of the policy, while this paper studies the estimation problem under more general metrics, which may be of high importance in real-world applications when people care about different functionals, such as CVaR in finance.

**Limitations And Societal Impact:**

Yes, they are discussed in Section 8.

**Main Review:**

Originality/Significance:

I like the problem this paper studies since it has some practical applicability and the method is easy to use with a solid statistical guarantee. I will view the significance level as high, as this more broad class of objectives characterizes lots of useful and popular metrics people care about. Regarding to the originality, I think it is a medium-low. The method proposed is based on CDF estimate and then a plug-in in any risk functionals, this kind of approach has been used in the previous literature as well, though not explicitly stated, see the algorithms [1]. Though this paper discusses its difference with a concurrent work [2], there is still some literature that utilizes this kind of approach [1], it would be great if the authors could comment on this.

Quality:

The quality of this paper is good. The motivation is clear, and the assumption of Lipschitz risk functional is well-motivated and justified. I like Section 4.1, but it might be shortened, and leave some space for experiments. The method is clearly introduced. Theoretically, this paper gives the first uniform concentration for CDF estimates in bandits, which also directly translates to a $O(1/\sqrt{n})$ rate for the Lipschitz objective. For the empirical study side, it seems all the examinations are about, first estimating CDF using different approaches, then a direct plug-in. I am wondering if there any other baselines in the literature that says do OPE for CVaR, variance, etc. If not, it would be great to state them clearly. If yes, it would be great to include them as baselines. I would view the current experiments as ablation studies, rather than a comparison with existing methods. Some specific comments:

[1]. How to estimate the Lipschitz constant when it is not obvious from the context? Or currently, it seems some estimates are not tight?
[2]. It seems the requirement of the Lipschitz property is to derive the uniform convergence rate theoretically. I am wondering how this method (probably empirically) applies to some metrics that not satisfy this requirement, say VaR? It would be great to comment on the necessity of this assumption.

[3]. Some typos:
line 6, it would be great to define CPT when use it the first time;
line 52, is scales;
line 35, off-hig evaluation;
line 90, steps generates;

Clarity:
I enjoy reading the paper and it writes well!


ref: [1]. Being Optimistic to Be Conservative: Quickly Learning a CVaR Policy
       [2]. Universal off-policy evaluation

**Time Spent Reviewing:**

3 hours

---

> ### Author Response · Authors · 2021-08-10
> **Response to Reviewer xxgN**
>
>
> $~$
>
> Thank you for your feedback, and we are glad to hear that you enjoyed reading our paper. We respond to your questions below:
>
> $~$
>
> **Re: Originality / Connection with Existing Literature**
>
> As the review correctly points out, several existing papers in the distributional reinforcement learning literature optimize risk functionals by first learning the distribution of rewards and then applying risk functionals on the distribution. In this sense, the approach of distribution + plugin risk estimate is not in itself novel. In fact, these papers provided some of the inspiration for our work.
>
> However prior to our work, such distributional strategies had not been introduced in the off-policy evaluation literature (aside from a concurrent paper [3]), no error rates had been proven for off-policy CDF estimation, and the class of Lipschitz risk functionals, for which errors in CDF estimation can be related to errors in risk estimation, had not been introduced. Our work introduces the first off-policy concentration results for both CDF and Lipschitz risk estimation.
>
> $~$
>
> **Re: Baselines**
>
> We agree that rigorous comparisons against baselines are important and believe that our paper can be improved by including a discussion of previously proposed estimators for specific risk functionals, which exist for both the mean [1,4,6] and variance [5]. Please note that prior to our work (and concurrent work [3] which independently developed the IS estimator), no estimators existed for off-policy estimation of the broader class of risk functionals addressed in our paper (not even for the widely studied CVaR).
>
> Theoretically, we can demonstrate that the OPRA estimator is equivalent to existing estimators of the mean and variance. This follows from the definition of expectation calculated via CDF:
> * mean: using the distortion risk formulation, with $g(x) = x$, we have $\rho(Z) = \int_0^D 1 - F(t) dt = \mathbb{E}(Z)$
>
> * variance: using the provided formula (L167), $\rho(Z) = 2\int_0^D t(1-F_Z(t))dt - \left(\int_0^D 1-F_Z(t)dt \right)^2 = \mathbb{E}(Z^2) - \mathbb{E}(Z)^2 = \mathbb{V}(Z)$
>
> We empirically verified this equivalence by running the IS/WIS/DR mean estimators [1] and variance estimator [5] on the PageBlocks dataset from Figure 1, with average absolute errors (over 500 repeats) shown below:
>
> | Metric | Estimator | n = 100 (e-3) | n = 924 (e-4) | n = 5472 (e-5) |
> | ---- | ---- | ---- | ---- | ---- |
> | Mean | IS [1] | 7.52 | 7.93 | 12.52 |
> | | OPRA IS | 4.38 | 4.65 | 7.11 |
> | | DR [1] | 1.63 | 2.34 | 3.39 |
> | | OPRA DR | 1.63 | 2.34 | 3.39 |
> | | DM [1] | 3.17 | 34.6 | 696 |
> | | OPRA DM | 3.17 | 34.6 | 696 |
> | Variance | IS [5] | 0.050 | 0.040 | 0.054 |
> | | OPRA IS | 0.007 | 0.007 | 0.009 |
> | | OPRA DR | 0.006 | 0.007 | 0.011 |
>
> Due to time constraints we were unable to run more complex estimators of the mean, such as [4], but will be happy to update our draft with these baselines in the revised version.
>
> $~$
>
> **Re: “How to estimate the Lipschitz constant when it is not obvious from the context? Or currently, it seems some estimates are not tight?”**
>
> For mean and CVaR, the Lipschitz constants given are tight. In other words, there exists cases for which the risk and CDF errors are related exactly by this value. Two complications can arise: (1) for some risk functionals, deriving the Lipschitz constant may not be easy, or we must settle for loose upper bounds; (2) while the Lipschitz constant gives us a distribution-independent upper bound on the risk (as a function of the CDF error), it’s possible that these constants may in general be pessimistic and we might hope to obtain tighter distribution-dependent bounds.
>
> $~$
>
> **Re: “I am wondering how this method (probably empirically) applies to some metrics that [do] not satisfy [Lipschitzness], say VaR?"**
>
> Absent any further assumptions, it is impossible to provide convergence rates on estimation of general metrics, such as the VaR, for any estimator. Consider for example, a case in which the reward takes value 0 in 20% of samples and value 1 in 80% of samples. Here no estimator of the VaR at level 0.2 will converge (it will oscillate between 0 and 1).
>
> If we additionally assume that the subgradient of the CDF $F$ is bounded below, we can guarantee that the VaR estimation converges at the same $O(1/\sqrt{n})$ rate as CDF estimation. However, this assumption may not always hold depending on the distribution underlying $F$. We prove this quickly below:
>
> >Let $t = F^{-1}(x)$ and $t' = \widehat{F}^{-1}(x)$. Further, let $x' = F(t')$, and assume WLOG that $t \geq t'$ so $x \geq x'$.
>
> >Denote $\partial F(t)$ to be the subgradient of $F$ at the point $t$, and suppose that the subgradient is bounded below $\partial F(t) \geq c$.
> >Then using the definition of the subgradient,
> $F(t) - F(t') \geq \partial F(t')(t-t') \geq c(t - t') = c\left( F^{-1}(x) - \widehat{F}^{-1}(x) \right) \geq 0$.
>
> >Using the identity that $F(t) = x = \widehat{F}(t')$, this implies that
> $\widehat{F}(t') - F(t') \geq c\left( F^{-1}(x) - \widehat{F}^{-1}(x) \right)$.
>
> >The LHS is exactly the CDF estimation error. If this error is small, i.e. $\delta \geq \widehat{F}(t') - F(t')$, we have
> $\frac{\delta}{c} \geq F^{-1}(x) - \widehat{F}^{-1}(x)$
> which implies the VaR estimation error is also small.
>
> The uniform concentrations in Theorems 5.1 and 5.2 thus give a concentration on VaR estimation error under assumption that the subgradient of $F$ is lower bounded.
>
> To summarize, although our method works for general risk functional estimation, derivation of error bounds for non-Lipschitz risk functionals is more nuanced. We thank the reviewer for raising this important point, and will include a detailed discussion in the revised version.
>
> $~$
>
> **Typos**
>
> Thank you very much for pointing these out to us. We will make all of these corrections in the camera-ready version.
>
> $~$
>
> **References**
>
> [1] Dudik et al. (2011) “Doubly Robust Policy Evaluation and Learning”
>
> [2] Wood and Zhang (1995) “Estimation of the Lipschitz Constant of a Function” https://link.springer.com/content/pdf/10.1007/BF00229304.pdf
>
> [3] Chandak et al. (2021) “Universal Off-Policy Evaluation”
>
> [4] Wang et al. (2017) “Optimal and Adaptive Off-policy Evaluation in Contextual Bandits”
>
> [5] Chandak et al. (2021) “High-Confidence Off-Policy (or Counterfactual) Variance Estimation”
>
> [6] Thomas et al. (2016) “Data-Efficient Off-Policy Policy Evaluation for Reinforcement Learning”

---

### Official Review · Reviewer_GLvQ · 2021-07-16

**Rating:** 5
**Confidence:** 4

**Summary:**

The paper studies the problem of risk assessment with offline data. Unlike traditional work that focuses on the offline evaluation of policies, this paper introduces a new class of risk functionals (of the CDF) as the estimands and generalizes the IS estimator and AIPW estimator for the estimation of the CDF (and the risk functionals).

**Limitations And Societal Impact:**

Discussion on potential negative societal impact should be added.

**Main Review:**

1. Novelty:

my main concern for this work is that it seems to be a straightforward combination of the well-known properties of the IS estimator and the AIPW estimator and a new class of estimands. The authors introduce a new class of estimands that are (Lipschitz) functionals of the CDF, so to estimate these estimands well it suffices to estimate the CDFs well. Establishing the concentration results for CDF estimates is also a straightforward adaptation of the results in the offline policy evaluation literature. Therefore, I feel there is a lack of technical novelty. Please correct me if I missed something.

2.  Stronger results:

Continuing with the above comment, I think the paper can be improved if stronger results can be shown. For example, is it possible to provide (asymptotic) confidence intervals for the estimands? Is it possible to establish optimality for the current results?

3. Minor:

Line 125: “its value” should be “the difference between the function values evaluated at Z and Z’”?

Line 158: the definition of u^+(z) should depend on c as well?

Line 351: might be improved


**Time Spent Reviewing:**

3 hours

---

> ### Author Response · Authors · 2021-08-10
> **Response to Reviewer GLvQ**
>
>
> $~$
>
> **Re: Novelty / “Establishing the concentration results for CDF estimates is also a straightforward adaptation of the results in the offline policy evaluation literature.”**
>
> We respectfully assert that our concentration results for CDF estimates are not adapted from any existing bounds in the off-policy evaluation (OPE) literature. Our results are uniform bounds on the distribution, meaning that they bound $\sup_t |F(t) - \widehat{F}(t)|$. By contrast, existing results in the OPE literature are pointwise bounds on individual risk functionals (such as the mean), so they can only be used to bound $|F(t) - \widehat{F}(t)|$ for a single $t$.
>
> In order to obtain uniform bounds, we took inspiration from the original proof of the DKW inequality [7], which bounds error in empirical CDF estimation (in the general statistics setting). This is a technically novel contribution and proof technique in the context of the OPE literature, and the first finite-sample uniform bound for off-policy CDF estimation and risk estimation.
>
> To highlight the technical novelty of our paper in the context of OPE literature, we point to a concurrent work [3] that also aims to estimate a broad class of risk functionals. Notably they do not establish uniform convergence of the CDF estimates or consistency of the downstream risk estimates, citing the development of a DKW-style inequality for off-policy estimation as an important research direction.
>
> $~$
>
> **Re: Strong Results / Asymptotic Confidence Intervals and Optimality**
>
> We provide exact confidence intervals that hold for all sample sizes $n$. While we have not derived the exact asymptotic variance of our estimators, we know that it cannot improve over our finite-sample rates. Notably, a recent work [5] demonstrated that the lower bound on off-policy mean estimation in multi-armed bandits is $O(R_{max}\sqrt{d/n})$ for large enough $n$, where $d$ corresponds to $w_2 = \mathbb{E}_{\mathbb{P}_\beta}[w^2]$, the exponent of second order Renyi divergence from our paper. Because the mean can be computed from the CDF, a lower bound on mean estimation entails a rate-matching lower bound on CDF estimation. Thus, our bounds are rate-optimal and match the asymptotic rates up to a constant factor.
>
> $~$
>
> **Minor Corrections**
>
> Thank you very much for bringing Lines 125, 158, and 351 to our attention. We will make all of these corrections in the camera-ready version.
>
> $~$
>
> **References**
>
> [1] Dudik et al. (2011) “Doubly Robust Policy Evaluation and Learning”
>
> [2] Chandak et al. (2021) “High-Confidence Off-Policy (or Counterfactual) Variance Estimation”
>
> [3] Chandak et al. (2021) “Universal Off-Policy Evaluation”
>
> [4] Wang et al. (2017) “Optimal and Adaptive Off-policy Evaluation in Contextual Bandits”
>
> [5] Ma et al. (2021) “Minimax Off-Policy Evaluation for Multi-Armed Bandits”
>
> [6] Li et al. (2014) “On Minimax Optimal Offline Policy Evaluation”
>
> [7] Dvoretzky (1956) “Asymptotic minimax character of the sample distribution function and of the classical multinomial estimator”

---

### Official Review · Reviewer_quRn · 2021-07-22

**Rating:** 8
**Confidence:** 4

**Summary:**

The paper proposes off-policy risk assessment(OPRA), a framework that estimate a risk functional
through first estimate CDF and second plugin estimates for any risks functional that
satisfies Lipschitz properties.
The new proposed framework is elegant and can simultaneously estimate a broad class
of objectives including CVar, variance, mean-variance and many others.
The paper also provides theoretical convergence analysis for the estimator.

**Limitations And Societal Impact:**

yes

**Main Review:**


I am satisfied with the novelty and the quality of the paper, and I think the paper is
well written and inspiring. I would recommend acceptance of the submission.

**Novelty:**
The paper provides the first DKW-style proof of uniform convergence of weighted empirical CDF
if the importance ratio is bounded.
The proof looks standard but is new and I think it is an important complement to
DKW-style inequalities.

**Quality:**
The proof is sound to me.
And I have the same concern of theorem 5.2 as the authors also mention in line 271.
The error for DR does not depend on $\hat{G} - G$ and cannot improve over IS also seems
weird to me.
A better bound on that would definitely improves the paper's quality.

**Clarity:**
The paper is well written and inspiring.
The background on Lipschitz risk functionals bring up a wide range of risk functionals
that we can directly utilized given the estimation of the CDF.
And the main proof in Section 5 discusses two important estimator: IS and DR,
both are provided with sufficient background the theoretical results.


**Minor:**
In line 229, 'DI' should replace with 'DM'.

**Time Spent Reviewing:**

3

---

> ### Author Response · Authors · 2021-08-10
> **Response to Reviewer quRn**
>
> Thank you for your feedback, and we are glad to hear that you found our paper inspiring to read. We hope to continue building upon our results in future work, and are actively investigating an improved error bound with the $\widehat{G}-G$ term.
>
> Our current bound is conservative in the sense that it upper bounds $\widehat{G} - G$ by its worst-case value, which is 1. This reflects scenarios where the model $\widehat{G}$ is poorly specified, and the DR estimator performs worse than the IS estimator. However, as you mentioned, inclusion of the $\widehat{G}-G$ term can quantify such misspecification, and will result in a smaller upper bound when the DR estimator does have better performance than IS.

---

### Official Review · Reviewer_8N59 · 2021-07-26

**Rating:** 7
**Confidence:** 4

**Summary:**

The work studies offline evaluation of policies assessed in terms of risk measures in a contextual bandit setting. It leverages the fact that a variety of risk measures are represented as some function of cumulative distribution function (CDF) of policy rewards with known Lipschitz constants. The work provides two consistent estimators of CDF, bounds their error from true CDF in sup norm, and use these to bound the risk measures. In experiments on two realistic datasets, the estimators are shown to have low error.

**Ethical Concerns:**

I do not foresee any ethics implications. Moving beyond mean rewards as the policy evaluation metric will provide more careful evaluation of policies that emphasizes extreme (possibly negative) rewards.

**Limitations And Societal Impact:**

Limitations of the analysis are mentioned. Please discuss limitations of the current experiments to evaluate methods across settings relevant to risk measures.

Suggestions to improve presentation:

Please add more detailed comparison of contributions between the concurrent work of Chandak et al. [9] and this work, highlighting the distinction between risk measures considered, guarantees for estimators (in more detail), importance of Lipschitz property to the results, and experimental comparison, if possible.

Please consider adding baselines specialised to estimating mean (e.g. https://www.cs.utexas.edu/~sniekum/classes/RL-F17/papers/HCOPE.pdf), variance, and CVaR. Is the CDF based approach competitive in terms of error and interval width?

Please consider plotting percent coverage and width of the confidence intervals. The intervals in Figures 1 and 2 for different methods are not easily distinguishable, so it hard to visually verify coverage and reduction in interval width with number of samples. For example, width of DR is not visible in Figure 2.

Please explicitly write the assumption making the problem a contextual bandit, i.e. independent draws from context distribution, in contrast to the full RL problem.

Please consider including experiments on datasets where risk measures-based evaluation is well motivated for example medical applications. Some potential datasets are antimicrobial resistance http://clinicalml.org/data/amr-dataset/, or vasopressor and fluid use in ICU https://github.com/matthieukomorowski/AI_Clinician, https://github.com/clinicalml/gumbel-max-scm).


Minor:

Line 35 ‘off-hig’

**Main Review:**

The work provides a general way of performing high-confidence off-policy value estimation which is applicable to a large class of evaluation metrics. These metrics, characterised as being Lipschitz functionals of CDF, are understudied in the machine learning literature and are practically relevant for policy evaluation in high-risk applications. The proposed approach that starts from bounding errors in estimating CDF and carrying this over to bounding errors in risk measure-based value is novel. As a result, the method guarantees good value estimates for any set of risk measures simultaneously, given they satisfy Lipschitz property. Experimental evaluation adequately verifies the theoretical results on realistic datasets.

The advantage of the approach is its generality for all Lipschitz risk measures. But familiarity with applications of such measures and their relative merits is assumed. Although many examples of such measures are presented (beyond mean, variance, and CVaR), some more details on strengths of each and applications in which they are appropriate will help to provide more guidance on use of such risk measures by the machine learning community.
Experimental evaluation does not seem to be suited to ideal settings where risk measures are useful, e.g. continuous-valued rewards with a skewed distribution and a large range (max-min is high). For binary rewards, as in the experiments, Lipschitz constant determined by max reward D is relatively small. While classification datasets are easier to convert to a bandit task, alternatives based on semi-synthetic datasets with simulated continuous-valued outcomes could be explored. This may provide understanding of usefulness of the confidence intervals when Lipschitz constants are large.


Questions I will like the authors to address:

Does the approach assume non-negative rewards (for applying Lemma 4.1)? Is this without any loss of generality?

How does the bounds from the presented approach compare with existing ones for mean, variance, and CVaR?

Does the experiments provide a sense of how tight are the confidence intervals for different risk measures and datasets?

In case of unknown behaviour policy, doubly robust estimators have another advantage (in addition to being consistent under misspecification of either action or reward models) of having root n convergence rates under mild misspecification. Does this property follows as well for the proposed doubly robust estimator, both for CDF and policy value?

**Time Spent Reviewing:**

6

---

> ### Author Response · Authors · 2021-08-10
> **Response to Reviewer 8N59**
>
>
> $~$
>
> **Re: “Does the approach assume non-negative rewards (for applying Lemma 4.1)? Is this without any loss of generality?”**
>
> We originally stated Lemma 4.1 using non-negative random variables on support $[0, D]$ because distortion risk functionals are conventionally defined for non-negative random variables [3]. However, this is WLOG and Lemmas 4.1-4.3 apply to any real rewards with bounded support. Thank you for raising this point. We will clarify the generality of our result in the revised version.
>
> $~$
>
> **Re: “How do the bounds from the presented approach compare with existing ones for mean, variance, and CVaR?”**
>
> While finite-sample error bounds exist for the off-policy estimation of the mean [5] and the variance [1], to our knowledge, our paper provides the first finite-sample bounds for CVaR. While [6] provide loose post-hoc bounds, they provide no convergence rates (or even proofs of consistency).
>
> In short, our analysis recovers the same rates as existing bounds for the mean and variance but also applies to the broader class of Lipschitz risk functionals. Below, we compare our bounds to pre-existing bounds specialized to the mean and variance, omitting discussion of constants for brevity. Here,  $d = \mathbb{E}_{\mathbb{P}_\beta}[w^2]$ denotes the average squared importance weight under the behavioral policy $\beta$, and $\hat{R}$ denotes the model of the reward under doubly robust mean estimation:
>
> | Metric | IS [5]  | OPRA IS | DR [1] | OPRA DR |
> | ---- | ---- | ---- | ---- | ---- |
> | Mean | $O\left(R_{max}\sqrt{\frac{d}{n}}\right)$ |  $O\left(R_{max}\sqrt{\frac{d}{n}}\right)$ |  $O\left(\sqrt{\frac{w_{max}^2 \mathbb{E}[(\widehat{R} - R)^2] + R_{max}^2\mathbb{V}_x[w] }{n}}\right)$ |  $O\left(R_{max}\sqrt{\frac{d}{n}}\right)$ |
> | Variance | $O\left(R^2_{max}\sqrt{\frac{d}{n}}\right)$ | $O\left(R^2_{max}\sqrt{\frac{d}{n}}\right)$ |  N/A | $O\left(R^2_{max}\sqrt{\frac{d}{n}}\right)$ |
>
> We give a brief justification. For the mean, the Lipschitz constant is $L = R_{max}$, or the maximum reward. Then our upper bounds translate to $O(R_{max}\sqrt{d/n})$. We note that, while the mean DR estimator bound includes the model error term $\widehat{R} - R$, our bound for the CDF DR estimator is conservative and upper bounds the model error with its maximum possible value. Incorporating the model error into the CDF DR error bound is an open question we continue to pursue.
>
> For the variance, the Lipschitz constant is $L = 3R_{max}^2$. Then our upper bound translates to $O(R_{max}^2\sqrt{d/n})$. The only existing bound (to our knowledge comes from [1], which for the random variable of rewards $R$ makes an off-policy estimate of the variance as $\mathbb{E}[wR^2] - \mathbb{E}[wR]^2$, where $w$ is the importance weight.
>
> $~$
>
> **Re: “Does the experiments provide a sense of how tight are the confidence intervals for different risk measures and datasets?”**
>
> As is typical for concentration inequalities of this nature, the error bounds (and thus confidence intervals) are relatively conservative for smaller $n$. In the camera-ready version, we will add comparisons of the theoretical vs empirical confidence intervals.
>
> From a theoretical perspective, a recent work [7] demonstrated that for large enough $n$, the lower bound on off-policy mean estimation in multi-armed bandits is $O(R_{max}\sqrt{d/n})$. Because CDF estimation boils down to estimation of the mean (of an indicator variable), a lower bound on mean estimation entails a rate-matching lower bound on CDF estimation, which our upper bounds match up to a constant. Thus, it is not possible to establish faster rates of convergence.
>
> $~$
>
> **Re: Advantage (root n convergence) of doubly robust estimators under misspecification**
>
> This is a great question and a difficult problem. We are actively working to derive error rates under misspecification but for now, this remains an open question.
>
> $~$
>
> **Suggestions**
>
> Thank you very much for your insightful suggestions, and we will incorporate all of them into the camera-ready version. We provide some preliminary results re: baselines, real-world datasets below.
>
> $~$
>
> **Re: Baselines specialized for estimating individual risk functionals**
>
> Thank you for pointing this out to us. We will update our draft with these baselines in the camera-ready version. We have included theoretical and numerical comparisons in our response to R4, who also shared this suggestion.
>
> $~$
>
> **Re: Real world-motivated datasets**
>
> Thank you very much for the linked datasets. While we did not have enough time to complete experiments with them in time for this response, we expect to have the results in time for the final version.
>
> We were more familiar with the Simglucose diabetes simulator [8], which also has continuous returns. Here, the policy controls the amount of insulin injection for diabetic patients. Each context corresponds to one of 4 different patients, and 5 actions correspond to the discretized space over possible injection amounts. Rewards are continuous on the range [-200, 200].
>
> We present preliminary results below, showing the sup-norm CDF estimation error and absolute risk estimation error (averaged over 100 trials). Note that in the results below, the doubly robust estimator outperforms the others in all estimation tasks:
>
> | Metric | Estimator | n = 200 | n = 400 | n = 1000 | n = 4000 |
> | ---- | ---- | ---- | ---- | ---- | ---- |
> |CDF | IS | 0.057 | 0.044 | 0.026 | 0.014 |
> |  | WIS | 0.057 | 0.044 | 0.026 | 0.014 |
> |  | DR | **0.048** | **0.034** | **0.021** | **0.011** |
> |  | DM | 0.049 | 0.035 | 0.021 | 0.011 |
> |Mean | IS | 3.440 | 2.728 | 1.575 | 0.823 |
> |  | WIS| 3.440 | 2.728 | 1.575 | 0.823 |
> |  | DR | **2.258** | **1.691** | 1.098 | **0.527** |
> |  | DM | 2.312 | 1.714 | **1.094** | 0.529 |
> | CVaR 0.25 | IS | 6.766 | 4.361 | 2.775 | 1.352 |
> |  | WIS| 3.440 |  6.766 | 4.361 | 2.775 | 1.352 |
> |  | DR | **3.457** | **2.612** | **1.494** | **0.789** |
> |  | DM | 3.729 | 2.651 | 1.481 | 0.793 |
>
> We will update our paper with more thorough results in the camera-ready version, and thank you again for this suggestion.
>
> $~$
>
> **References**
>
> [1] Chandak et al. (2021) “High-Confidence Off-Policy (or Counterfactual) Variance Estimation”
>
> [2] Thomas et al. (2015) “High Confidence Off-Policy Evaluation”
>
> [3] Balbas et al. (2009) “Properties of Distortion Risk Measures”
>
> [4] Prashanth et al. (2016) “Cumulative prospect theory meets reinforcement learning: Prediction and control”
>
> [5] Dudik et al. (2011) “Doubly Robust Policy Evaluation and Learning”
>
> [6] Chandak et al. (2021) “Universal Off-Policy Evaluation”
>
> [7] Ma et al. (2021) “Minimax Off-Policy Evaluation for Multi-Armed Bandits”
>
> [8] Xi (2018) Simglucose v0.2.1. https://github.com/jxx123/simglucose

---

> > ### Comment · Reviewer_8N59 · 2021-08-25
> > **Thanks for the response**
> >
> > My questions have been adequately addressed. I thank the authors for the detailed response and encourage them to add the results they presented here (and promise to add) in the final draft, specifically, the Simglucose results and comparison of error bounds for mean, variance, CVaR from existing results. I retain my evaluation score of Accept.

---

### Author Response · Authors · 2021-08-10
**General Reply to All Reviewers**

We would like to thank all of our reviewers for thoughtful and constructive feedback. We are glad to see that 3 reviewers recommend acceptance, and that the reviewers found our paper to be "well-written and inspiring" (R2), with "high significance level" (R4), "solid statistical guarantees" (R4), and "realistic experiments" (R1). We are also grateful to the reviewers for providing constructive suggestions and clarification questions. Please find our responses to each reviewer’s concerns in the respective threads.

---

### Decision · Program_Chairs · 2021-09-27

**Decision:**

Accept (Poster)

**Comment:**

This paper considers the off-policy Lipschitz risk functional estimation in contextual bandit setting. The authors considered plug-in estimator based on the CDF-estimator, with guarantees from DKW-inequality. The paper is well-written: the motivation is clear and the logic flow is easy to follow.  The method is also novel.

As the reviewers suggested, the experiment part is relatively weak. Please consider to add the extra empirical study into final version.

Another issue is that there are some important related work missing, e.g., [1, 2]. Therefore, the method is not well-positioned in literature. Please consider the comparison with [1, 2] and add the table in the response to Reviewer 8N59.

[1] Faury, Louis, Ugo Tanielian, Elvis Dohmatob, Elena Smirnova, and Flavian Vasile. "Distributionally robust counterfactual risk minimization." In Proceedings of the AAAI Conference on Artificial Intelligence, vol. 34, no. 04, pp. 3850-3857. 2020.

[2] Karampatziakis, Nikos, John Langford, and Paul Mineiro. "Empirical likelihood for contextual bandits." arXiv preprint arXiv:1906.03323 (2019).